# On the Equivalence between Neural Network and Support Vector Machine

**Yilan Chen**
Computer Science and Engineering
University of California San Diego
La Jolla, CA
`yilan@ucsd.edu`

**Wei Huang**
Engineering and Information Technology
University of Technology Sydney
Ultimo, Australia
`weihuang.uts@gmail.com`

**Lam M. Nguyen**
IBM Research
Thomas J. Watson Research Center
Yorktown Heights, NY
`LamNguyen.MLTD@ibm.com`

**Tsui-Wei Weng**
Halıcıoğlu Data Science Institute
University of California San Diego
La Jolla, CA
`lweng@ucsd.edu`

## Abstract

Recent research shows that the dynamics of an infinitely wide neural network (NN) trained by gradient descent can be characterized by Neural Tangent Kernel (NTK) [27]. Under the squared loss, the infinite-width NN trained by gradient descent with an infinitely small learning rate is equivalent to kernel regression with NTK [4]. However, the equivalence is only known for ridge regression currently [6], while the equivalence between NN and other kernel machines (KMs), e.g. support vector machine (SVM), remains unknown. Therefore, in this work, we propose to establish the equivalence between NN and SVM, and specifically, the infinitely wide NN trained by soft margin loss and the standard soft margin SVM with NTK trained by subgradient descent. Our main theoretical results include establishing the equivalence between NN and a broad family of $\ell_2$ regularized KMs with finite-width bounds, which cannot be handled by prior work, and showing that every finite-width NN trained by such regularized loss functions is approximately a KM. Furthermore, we demonstrate our theory can enable three practical applications, including (i) *non-vacuous* generalization bound of NN via the corresponding KM; (ii) *nontrivial* robustness certificate for the infinite-width NN (while existing robustness verification methods would provide vacuous bounds); (iii) intrinsically more robust infinite-width NNs than those from previous kernel regression.

## 1 Introduction

Recent research has made some progress towards deep learning theory from the perspective of infinite-width NN. For a fully-trained neural network, it follows kernel gradient descent in the function space with respect to NTK [27]. Under this linear regime and squared loss, it is rigorously proved that the fully-trained net is equivalent to kernel regression with NTK [4], which gives the generalization ability of such a model [5]. NTK helps us understand the optimization [27, 18] and generalization [5, 12] of NN through the perspective of kernels. However, existing theories about NTK [27, 30, 4, 13] usually assume the loss is a function of the model output, which does not include the case of regularization. Besides, they usually consider the squared loss which corresponds to a kernel regression, which may have limited insights to understand classification problems since squared loss and kernel regression are usually used for regression problems.

35th Conference on Neural Information Processing Systems (NeurIPS 2021).

On the other hand, another popular machine learning paradigm with solid theoretical foundation before the prevalence of deep neural networks is the support vector machine (SVM) [10, 15], which allows learning linear classifiers in high dimensional feature spaces. SVM tackles the sample complexity challenge by searching for large margin separators and tackles the computational complexity challenge using the idea of kernels [43]. To learn an SVM model, it usually involves solving a dual problem which is cast as a convex quadratic programming problem. Recently, there are some algorithms using subgradient descent [44] and coordinate descent [23] to further scale the SVM models to large datasets and high dimensional feature spaces.

We noticed that existing theoretical analysis mostly focused on connecting NN with kernel regression [27, 4, 30] but the connections between NN and SVM have not yet been explored. In this work, we establish the equivalence between NN and SVM for the first time to our best knowledge. More broadly, we show that our analysis can connect NNs with a family of $\ell_2$ regularized KMs, including kernel ridge regression (KRR), support vector regression (SVR) and $\ell_2$ regularized logistic regression, where previous results [27, 4, 30] cannot handle. These are the equivalences beyond ridge regression for the first time. Importantly, the equivalence between infinite-width NN and these $\ell_2$ regularized KMs may shed light on the understanding of NN from these new equivalent KMs [16, 45, 42, 48], especially towards understanding the training, generalization, and robustness of NN for classification problems. Besides, regularization plays an important role in machine learning to restrict the complexity of models. This equivalence may shed light on the understanding of the regularization for NN. We highlight our contributions as follows:

- We derive the continuous (gradient flow) and discrete dynamics of SVM trained by subgradient descent and the dynamics of NN trained by soft margin loss. We show the dynamics of SVM with NTK and NN are exactly the same in the infinite width limit because of the constancy of the tangent kernel and thus establish the equivalence. We show same linear convergence rate of SVM and NN under reasonable assumption. We verify the equivalence by experiments of subgradient descent and stochastic subgradient descent on MNIST dataset [28].

- We generalize our theory to general loss functions with $\ell_2$ regularization and establish the equivalence between NN and a family of $\ell_2$ regularized KMs as summarized in Table 1. We prove the difference between the outputs of SVM and NN sacles as $O(\ln m/\lambda\sqrt{m})$, where $\lambda$ is the coefficient of the regularization and $m$ is the width of the NN. Additionally, we show every finite-width neural network trained by a $\ell_2$ regularized loss function is approximately a KM.

- We show that our theory offers three practical benefits: (i) computing *non-vacuous* generalization bound of NN via the corresponding KM; (ii) we can deliver *nontrivial* robustness certificate for the over-parameterized NN (with width $m \to \infty$) while existing robustness verification methods would give trivial robustness certificate due to bound propagation [22, 52, 55]. In particular, the certificate decreases at a rate of $O(1/\sqrt{m})$ as the width of NN increases; (iii) we show that the equivalent infinite-width NNs trained from our $\ell_2$ regularized KMs are more robust than the equivalent NN trained from previous kernel regression [27, 4] (see Table 3), which is perhaps not too surprising as the regularization has a strong connection to robust machine learning.

## 2 Related Works and Background

### 2.1 Related Works

**Neural Tangent Kernel and dynamics of neural networks**. NTK was first introduced in [27] and extended to Convolutional NTK [4] and Graph NTK [20]. [26] studied the NTK of orthogonal initialization. [6] reported strong performance of NTK on small-data tasks both for kernel regression and kernel SVM. However, the equivalence is only known for ridge regression currently, but not for SVM and other KMs. A line of recent work [19, 1] proved the convergence of (convolutional) neural networks with large but finite width in a non-asymptotic way by showing the weights do not move far away from initialization in the optimization dynamics (trajectory). [30] showed the dynamics of wide neural networks are governed by a linear model of first-order Taylor expansion around its initial parameters. However, existing theory about NTK [27, 30, 4] usually assume the loss is a function of the model output, which does not include the case of regularization. Besides, they usually consider the squared loss which corresponds to a kernel regression, which may have limited insights to understand classification problems since squared loss and kernel regression are usually

used for regression problems. In this paper, we study the regularized loss functions and establish the equivalence with KMs beyond kernel regression and regression problems.

Besides, we studied the robustness of NTK models. [24] studied the label noise (the labels are generated by a ground truth function plus a Gaussian noise) while we consider the robustness of input perturbation. They study the convergence rate of NN trained by $\ell_2$ regularized squared loss to an underlying true function, while we give explicit robustness certificates for NNs. Our robustness certificate enables us to compare different models and show the equivalent infinite-width NNs trained from our $\ell_2$ regularized KMs are more robust than the equivalent NN trained from previous kernel regression.

**Neural network and support vector machine**. Prior works [50, 49, 34, 47, 31] have explored the benefits of encouraging large margin in the context of deep networks. [14] introduced a new family of positive-definite kernel functions that mimic the computation in multilayer neural nets and applied the kernels into SVM. [17] showed that neural networks trained by gradient flow are approximately KMs with a new conceptual kernel named path kernel. [44] proposed a subgradient algorithm to solve the primal problem of SVM, which can obtain a solution of accuracy $\epsilon$ in $\tilde{O}(1/\epsilon)$ iterations, where $\tilde{O}$ omits the logarithmic factors. In this paper, we also consider the SVM trained by subgradient descent and connect it with NN trained by subgradient descent. [46, 3] studied the connection between SVM and regularization neural network [41], one-hidden layer NN that has very similar structures with that of KMs and is not widely used in practice. NNs used in practice now (e.g. fully connected ReLU NN, CNN, ResNet) do not have such structures. [40] analyzed NN trained by two-layer NN trained by hinge loss without regularization on linearly separable dataset. Note for SVM, it must have a regularization term such that it can achieve max-margin solution.

## 2.2 Neural Networks and Tangent Kernel

We consider a general form of deep neural network $f$ with a linear output layer as [32]. Let $[L] = \{1, ..., L\}, \forall l \in [L]$,

$$\alpha^{(0)}(w, x) = x, \ \alpha^{(l)}(w, x) = \phi_l(w^{(l)}, \alpha^{(l-1)}), \ f(w, x) = \frac{1}{\sqrt{m_L}} \langle w^{(L+1)}, \alpha^{(L)}(w, x) \rangle, \quad (1)$$

where each vector-valued function $\phi_l(w^{(l)}, \cdot) : \mathbb{R}^{m_{l-1}} \to \mathbb{R}^{m_l}$, with parameter $w^{(l)} \in \mathbb{R}^{p_l}$ ($p_l$ is the number of parameters), is considered as a layer of the network. This definition includes the standard fully connected, convolutional (CNN), and residual (ResNet) neural networks as special cases. For a fully connected ReLU NN, $\alpha^{(l)}(w, x) = \sigma(\frac{1}{\sqrt{m_{l-1}}} w^{(l)} \alpha^{(l-1)})$ with $w^{(l)} \in \mathbb{R}^{m_l \times m_{l-1}}$ and $\sigma(z) = \max(0, z)$.

**Initialization and parameterization.** In this paper, we consider the NTK parameterization [27], under which the constancy of the tangent kernel has been initially observed. Specifically, the parameters, $w := \{w^{(1)}; w^{(2)}; \cdots; w^{(L)}; w^{(L+1)}\}$ are drawn i.i.d. from a standard Gaussian, $\mathcal{N}(0, 1)$, at initialization, denoted as $w_0$. The factor $1/\sqrt{m_L}$ in the output layer is required by the NTK parameterization in order that the output $f$ is of order $O(1)$. While we only consider NTK parameterization here, the results should be able to extend to general parameterization of kernel regime [53].

**Definition 2.1** (Tangent Kernel). The tangent kernel associated with function $f(w, x)$ at some parameter $w$ is $\hat{\Theta}(w; x, x') = \langle \nabla_w f(w, x), \nabla_w f(w, x') \rangle$. Under certain conditions (usually infinite width limit and NTK parameterization), the tangent kernel at initialization converges in probability to a deterministic limit and keeps constant during training, $\hat{\Theta}(w; x, x') \to \Theta_\infty(x, x')$. This limiting kernel is called *Neural Tangent Kernel (NTK)*.

## 2.3 Kernel Machines

Kernel machine (KM) is a model of the form $g(\beta, x) = \varphi(\langle \beta, \Phi(x) \rangle + b)$, where $\beta$ is the model parameter and $\Phi$ is a mapping from input space to some feature space, $\Phi : \mathcal{X} \to \mathcal{F}$. $\varphi$ is an optional nonlinear function, such as identity mapping for kernel regression and $sign(\cdot)$ for SVM and logistic regression. The kernel can be exploited whenever the weight vector can be expressed as a linear combination of the training points, $\beta = \sum_{i=1}^{n} \alpha_i \Phi(x_i)$ for some value of $\alpha_i, i \in [n]$, implying that we can express $g$ as $g(x) = \varphi(\sum_{i=1}^{n} \alpha_i K(x, x_i) + b)$, where $K(x, x_i) = \langle \Phi(x), \Phi(x_i) \rangle$ is the kernel function. For a neural network in NTK regime, we have $f(w_t, x) \approx f(w_0, x) + \langle \nabla_w f(w_0, x), w_t - $

$w_0\rangle$, which makes the neural network linear in the gradient feature mapping $x \to \nabla_w f(w_0, x)$. Under squared loss, it is equivalent to kernel regression with $\Phi(x) = \nabla_w f(w_0, x)$ (or equivalently using NTK as the kernel), $\beta = w_t - w_0$ and $\varphi$ identity mapping [4].

As far as we know, there is no work establishing the equivalence between fully trained networks and SVM. [17] showed that neural networks trained by gradient flow are approximately KMs, but didn't discuss any specific KM. In this work, we compare the dynamics of SVM and neural network trained by subgradient descent with soft margin loss and show the equivalence between them in the infinite width limit.

## 2.4 Subgradient Optimization of Support Vector Machine

We first formally define the standard soft margin SVM and then show how the subgradient descent can be applied to get an estimation of the SVM primal problem. For simplicity, we consider the homogenous model, $g(\beta, x) = \langle \beta, \Phi(x) \rangle$.[1]

**Definition 2.2** (Soft Margin SVM). Given labeled samples $\{(x_i, y_i)\}_{i=1}^n$ with $y_i \in \{-1, +1\}$, the hyperplane $\beta^*$ that solves the below optimization problem realizes the soft margin classifier with geometric margin $\gamma = 2/\|\beta^*\|$.

$$\min_{\beta, \xi} \frac{1}{2}\|\beta\|^2 + C\sum_{i=1}^n \xi_i, \quad s.t.\ y_i\langle \beta, \Phi(x_i)\rangle \geq 1 - \xi_i,\ \xi_i \geq 0,\ i \in [n],$$

**Proposition 2.1.** *The above primal problem of soft margin SVM can be equivalently formulated as*

$$\min_{\beta} \frac{1}{2}\|\beta\|^2 + C\sum_{i=1}^n \max(0, 1 - y_i\langle \beta, \Phi(x_i)\rangle), \tag{2}$$

*where the second term is a hinge loss. Denote this function as $L(\beta)$, which is strongly convex in $\beta$.*

From this, we see that the SVM technique is equivalent to empirical risk minimization with $\ell_2$ regularization, where in this case the loss function is the nonsmooth hinge loss. The classical approaches usually consider the dual problem of SVM and solve it as a quadratic programming problem. Some recent algorithms, however, use subgradient descent [44] to optimize Eq. (2), which shows significant advantages when dealing with large datasets.

In this paper, we consider the soft margin SVM trained by subgradient descent with $L(\beta)$. We use the subgradient $\nabla_\beta L(\beta) = \beta - C\sum_{i=1}^n \mathbb{1}(y_i g(\beta, x_i) < 1)y_i\Phi(x_i)$, where $\mathbb{1}(\cdot)$ is the indicator function. As proved in [44], we can find a solution of accuracy $\epsilon$, i.e. $L(\beta) - L(\beta^*) \leq \epsilon$, in $\tilde{O}(1/\epsilon)$ iterations. Other works also give convergence guarantees for subgradient descent of convex functions [11, 9]. In the following analysis, we will generally assume the convergence of SVM trained by subgradient descent.

## 3 Main Theoretical Results

In this section, we describe our main results. We first derive the continuous (gradient flow) and discrete dynamics of SVM trained by subgradient descent (in Section 3.1) and the dynamics of NN trained by soft margin loss (in Section 3.2 and Section 3.3). We show that they have similar dynamics, characterized by an inhomogeneous linear differential (difference) equation, and have the same convergence rate under reasonable assumption. Next, we show that their dynamics are exactly the same in the infinite width limit because of the constancy of tangent kernel and thus establish the equivalence (Theorem 3.4). Furthermore, in Section 3.4, we generalize our theory to general loss functions with $\ell_2$ regularization and establish the equivalence between NN and a family of $\ell_2$ regularized KMs as summarized in Table 1.

### 3.1 Dynamics of Soft Margin SVM

For simpicity, we denote $\beta_t$ as $\beta$ at some time $t$ and $g_t(x) = g(\beta_t, x)$. The proofs of the following two theorems are detailed in Appendix C.

---

[1]Note one can always deal with the bias term $b$ by adding each sample with an additional dimension, $\Phi(x)^T \leftarrow [\Phi(x)^T, 1], \beta^T \leftarrow [\beta^T, 1]$.

**Theorem 3.1** (Continuous Dynamics and Convergence Rate of SVM). *Consider training soft margin SVM by subgradient descent with infinite small learning rate (gradient flow [2]): $\frac{d\beta_t}{dt} = -\nabla_\beta L(\beta_t)$, the model $g_t(x)$ follows the below evolution:*

$$\frac{dg_t(x)}{dt} = -g_t(x) + C\sum_{i=1}^{n} \mathbb{1}(y_i g_t(x_i) < 1) y_i K(x, x_i), \tag{3}$$

*and has a linear convergence rate:*

$$L(\beta_t) - L(\beta^*) \le e^{-2t}\left(L(\beta_0) - L(\beta^*)\right).$$

*Denote $Q(t) = C\sum_{i=1}^{n} \mathbb{1}(y_i g_t(x_i) < 1) y_i K(x, x_i)$, which changes over time until convergence. The model output $g_t(x)$ at some time $T$ is*

$$g_T(x) = e^{-T}\left(g_0(x) + \int_0^T Q(t)e^t\, dt\right), \quad \lim_{T\to\infty} g_T(x) = C\sum_{i=1}^{n} \mathbb{1}(y_i g_T(x_i) < 1) y_i K(x, x_i). \tag{4}$$

The continuous dynamics of SVM is described by an inhomogeneous linear differential equation (Eq. (3)), which gives an analytical solution. From Eq. (4), we can see that the influence of initial model $g_0(x)$ deceases as time $T \to \infty$ and disappears at last.

**Theorem 3.2** (Discrete Dynamics of SVM). *Let $\eta \in (0,1)$ be the learning rate. The dynamics of subgradient descent is*

$$g_{t+1}(x) - g_t(x) = -\eta g_t(x) + \eta C\sum_{i=1}^{n} \mathbb{1}(y_i g_t(x_i) < 1) y_i K(x, x_i). \tag{5}$$

*Denote $Q(t) = \eta C\sum_{i=1}^{n} \mathbb{1}(y_i g_t(x_i) < 1) y_i K(x, x_i)$, which changes over time. The model output $g_t(x)$ at some time $T$ is*

$$g_T(x) = (1-\eta)^T\left(g_0(x) + \sum_{t=0}^{T-1}(1-\eta)^{-t-1}Q(t)\right), \lim_{T\to\infty} g_T(x) = C\sum_{i=1}^{n} \mathbb{1}(y_i g_T(x_i) < 1) y_i K(x, x_i).$$

The discrete dynamics is characterized by an inhomogeneous linear difference equation (Eq. (5)). The discrete dynamics and solution of SVM have similar structures as the continuous case.

## 3.2 Soft Margin Neural Network

We first formally define the soft margin neural network and then derive the dynamics of training a neural network by subgradient descent with soft margin loss. We will consider a neural network defined as Eq. (1). For convenience, we redefine $f(w, x) = \langle W^{(L+1)}, \alpha^{(L)}(w, x)\rangle$ with $W^{(L+1)} = \frac{1}{\sqrt{m_L}}w^{(L+1)}$ and $w := \{w^{(1)}; w^{(2)}; \cdots; w^{(L)}; W^{(L+1)}\}$.

**Definition 3.1** (Soft Margin Neural Network). Given samples $\{(x_i, y_i)\}_{i=1}^{n}$, $y_i \in \{-1, +1\}$, the neural network $w^*$ defined as Eq. (1) that solves the following two equivalent optimization problems

$$\min_{w,\xi} \frac{1}{2}\|W^{(L+1)}\|^2 + C\sum_{i=1}^{n}\xi_i, \quad s.t.\ y_i f(w, x_i) \ge 1 - \xi_i,\ \xi_i \ge 0,\ i \in [n],$$

$$\min_{w} \frac{1}{2}\|W^{(L+1)}\|^2 + C\sum_{i=1}^{n}\max(0, 1 - y_i f(w, x_i)), \tag{6}$$

realizes the soft margin classifier with geometric margin $\gamma = 2/\|W_*^{(L+1)}\|$. Denote Eq. (6) as $L(w)$ and call it *soft margin loss*.

This is generally a hard nonconvex optimization problem, but we can apply subgradient descent to optimize it heuristically. At initilization, $\|W_0^{(L+1)}\|^2 = O(1)$. The derivative of the regularization for $w^{(L+1)}$ is $w^{(L+1)}/\sqrt{m_L} = O(1/\sqrt{m_L}) \to 0$. For a fixed $\alpha^{(L)}(w, x)$, this problem is same as SVM with $\Phi(x) = \alpha^{(L)}(w, x)$, kernel $K(x, x') = \alpha^{(L)}(w, x)\cdot\alpha^{(L)}(w, x')$ and parameter $\beta = W^{(L+1)}$. If we only train the last layer of NN, it corresponds to an SVM with a NNGP kernel [29, 36]. But for a fully-trained NN, $\alpha^{(L)}(w, x)$ is changing over time.

### 3.3 Dynamics of Neural Network Trained by Soft Margin Loss

Denote the hinge loss in $L(w)$ as $L_h(y_i, f(w, x_i)) = C \max(0, 1 - y_i f(w, x_i))$. We use the same subgradient as that for SVM, $L'_h(y_i, f(w, x_i)) = -Cy_i \mathbb{1}(y_i f(w, x_i) < 1)$.

**Theorem 3.3** (Continuous Dynamics and Convergence Rate of NN). *Suppose an NN $f(w, x)$ defined as Eq. (1), with $f$ a differentiable function of $w$, is learned from a training set $\{(x_i, y_i)\}_{i=1}^n$ by subgradient descent with $L(w)$ and gradient flow. Then the network has the following dynamics:*

$$\frac{df_t(x)}{dt} = -f_t(x) + C \sum_{i=1}^n \mathbb{1}(y_i f_t(x_i) < 1) y_i \hat{\Theta}(w_t; x, x_i).$$

*Let $\hat{\Theta}(w_t) \in \mathbb{R}^{n \times n}$ be the tangent kernel evaluated on the training set and $\lambda_{min}(\hat{\Theta}(w_t))$ be its minimum eigenvalue. Assume $\lambda_{min}(\hat{\Theta}(w_t)) \geq \frac{2}{C}$, then NN has at least a linear convergence rate, same as SVM:*

$$L(w_t) - L(w^*) \leq e^{-2t} \left( L(w_0) - L(w^*) \right).$$

The proof is in Appendix D. The key observation is that when deriving the dynamics of $f_t(x)$, the $\frac{1}{2}\|W^{(L+1)}\|^2$ term in the loss function will produce a $f_t(x)$ term and the hinge loss will produce the tangent kernel term, which overall gives a similar dynamics to that of SVM. Comparing to the previous continuous-time gradient descent [27, 30], our result has an extra $-f_t(x)$ here because of the regularization term of the loss function. The convergence rate is proved based on a sufficient condition for the PL inequality. The assumption of $\lambda_{min}(\hat{\Theta}(w_t)) \geq \frac{2}{C}$ can be guaranteed in a parameter ball when $\lambda_{min}(\hat{\Theta}(w_0)) > \frac{2}{C}$, by using a sufficiently wide NN [33].

If the tangent kernel $\hat{\Theta}(w_t; x, x_i)$ is fixed, $\hat{\Theta}(w_t; x, x_i) \to \hat{\Theta}(w_0; x, x_i)$, the dynamics of NN is the same as that of SVM (Eq. (3)) with kernel $\hat{\Theta}(w_0; x, x_i)$, assuming the neural network and SVM have same initial output $g_0(x) = f_0(x)$.[2] And this consistency of tangent kernel is the case for infinitely wide neural networks of common architectures, which does not depend on optimization algorithm and the choice of loss function, as discussed in [32].

**Assumptions.** We assume that (vector-valued) layer functions $\phi_l(w, \alpha), l \in [L]$ are $L_\phi$-Lipschitz continuous and twice differentiable with respect to input $\alpha$ and parameters $w$. The assumptions serve for the following theorem to show the constancy of tangent kernel.

**Theorem 3.4** (Equivalence between NN and SVM). *As the minimum width of the NN, $m = \min_{l \in [L]} m_l$, goes to infinity, the tangent kernel tends to be constant, $\hat{\Theta}(w_t; x, x_i) \to \hat{\Theta}(w_0; x, x_i)$. Assume $g_0(x) = f_0(x)$. Then the infinitely wide NN trained by subgradient descent with soft margin loss has the same dynamics as SVM with $\hat{\Theta}(w_0; x, x_i)$ trained by subgradient descent:*

$$\frac{df_t(x)}{dt} = -f_t(x) + C \sum_{i=1}^n \mathbb{1}(y_i f_t(x_i) < 1) y_i \hat{\Theta}(w_0; x, x_i).$$

*And thus such NN and SVM converge to the same solution.*

The proof is in Appendix E. We apply the results of [32] to show the constancy of tangent kernel in the infinite width limit. Then it is easy to check the dynamics of infinitely wide NN and SVM with NTK are the same. We give a finite-width bound for general loss functions in the next section. This theorem establishes the equivalence between infinitely wide NN and SVM for the first time. Previous theoretical results of SVM [16, 45, 42, 48] can be directly applied to understand the generalization of NN trained by soft margin loss. Given the tangent kernel is constant or equivalently the model is linear, we can also give the discrete dynamics of NN (Appendix D.4), which is identical to that of SVM. Compared with the previous discrete-time gradient descent [30, 53], our result has an extra $-\eta f_t(x)$ term because of the regularization term of loss function.

$$f_{t+1}(x) - f_t(x) = -\eta f_t(x) + \eta C \sum_{i=1}^n \mathbb{1}(y_i f_t(x_i) < 1) y_i \hat{\Theta}(w_0; x, x_i).$$

---

[2]This can be done by setting the initial values to be 0, i.e. $g_0(x) = f_0(x) = 0$.

Table 1: Summary of our theoretical results on the equivalence between infinite-width NNs and a family of KMs. Thanks to the representer theorem [42], our $\ell_2$ regularized KMs can all apply kernel trick, meaning infinite NTK can be applied in these $\ell_2$ regularized KMs.

| $\lambda$ | Loss $l(z, y_i)$ | Kernel machine |
|---|---|---|
| $\lambda = 0$ ([27], [4]) | $(y_i - z)^2$ | Kernel regression |
| $\lambda \to 0$ (ours) | $\max(0, 1 - y_i z)$ | Hard margin SVM |
| $\lambda > 0$ (ours) | $\max(0, 1 - y_i z)$ | (1-norm) soft margin SVM |
| | $\max(0, 1 - y_i z)^2$ | 2-norm soft margin SVM |
| | $\max(0, \|y_i - z\| - \epsilon)$ | Support vector regression |
| | $(y_i - z)^2$ | Kernel ridge regression (KRR) |
| | $\log(1 + e^{-y_i z})$ | Logistic regression with $\ell_2$ regularization |

### 3.4 General Loss Functions

We note that above analysis does not have specific dependence on the hinge loss. Thus we can generalize our analysis to general loss functions $l(z, y_i)$, where $z$ is the model output, as long as the loss function is differentiable (or has subgradients) with respect to $z$, such as squared loss and logistic loss. Besides, we can scale the regularization term by a factor $\lambda$ instead of scaling $l(z, y_i)$ with $C$ as it for SVM, which are equivalent. Suppose the loss function for the KM and NN are

$$L(\beta) = \frac{\lambda}{2}\|\beta\|^2 + \sum_{i=1}^{n} l(g(\beta, x_i), y_i), \quad L(w) = \frac{\lambda}{2}\|W^{(L+1)}\|^2 + \sum_{i=1}^{n} l(f(w, x_i), y_i). \quad (7)$$

Then the continuous dynamics of $g_t(x)$ and $f_t(x)$ are

$$\frac{dg_t(x)}{dt} = -\lambda g_t(x) - \sum_{i=1}^{n} l'(g_t(x_i), y_i) K(x, x_i), \quad (8)$$

$$\frac{df_t(x)}{dt} = -\lambda f_t(x) - \sum_{i=1}^{n} l'(f_t(x_i), y_i) \hat{\Theta}(w_t; x, x_i), \quad (9)$$

where $l'(z, y_i) = \frac{\partial l(z, y_i)}{\partial z}$. In the situation of $\hat{\Theta}(w_t; x, x_i) \to \hat{\Theta}(w_0; x, x_i)$ and $K(x, x_i) = \hat{\Theta}(w_0; x, x_i)$, these two dynamics are the same (assuming $g_0(x) = f_0(x)$). When $\lambda = 0$, we recover the previous results of kernel regression. When $\lambda > 0$, we have our new results of $\ell_2$ regularized loss functions. Table 1 lists the different loss functions and the corresponding KMs that infinite-width NNs are equivalent to. KRR is considered in [25] to analyze the generalization of NN. However, they directly assume NN as a linear model and use it in KRR. Below we give finite-width bounds on the difference between the outputs of NN and the corresponding KM. The proof is in F.

**Theorem 3.5** (Bounds on the difference between NN and KM). *Assume $g_0(x) = f_0(x), \forall x$ and $K(x, x_i) = \hat{\Theta}(w_0; x, x_i)$ [3]. Suppose the KM and NN are trained with losses (7) and gradient flow. Suppose $l$ is $\rho$-lipschitz and $\beta_l$-smooth for the first argument (i.e. the model output). Given any $w_T \in B(w_0; R) := \{w : \|w - w_0\| \le R\}$ for some fixed $R > 0$, for training data $X \in \mathbb{R}^{d \times n}$ and a test point $x \in \mathbb{R}^d$, with high probability over the initialization,*

$$\|f_T(X) - g_T(X)\| = O(\frac{e^{\beta_l \|\hat{\Theta}(w_0)\|} R^{3L+1} \rho n^{\frac{3}{2}} \ln m}{\lambda \sqrt{m}}),$$

$$\|f_T(x) - g_T(x)\| = O(\frac{e^{\beta_l \|\hat{\Theta}(w_0; X, x)\|} R^{3L+1} \rho n \ln m}{\lambda \sqrt{m}}).$$

*where $f_T(X), g_T(X) \in \mathbb{R}^n$ are the outputs of the training data and $\hat{\Theta}(w_0; X, x) \in \mathbb{R}^n$ is the tangent kernel evaluated between training data and test point.*

---

[3]Linearized NN is a special case of such $g$.

# 4 Discussion

In this section, we give some extensions and applications of our theory. We first show that every finite-width neural network trained by a $\ell_2$ regularized loss function is approximately a KM in Section 4.1, which enables us to compute non-vacuous generalization bound of NN vis the corresponding KM. Next, in Section 4.2, we show that our theory of equivalence (in Section 3.3) is useful to evaluating the robustness of over-parameterized NNs with infinite width. In particular, our theory allows us to deliver nontrivial robustness certificates for infinite-width NNs, while existing robustness verification methods [22, 52, 55] would become much looser (decrease at a rate of $O(1/\sqrt{m})$) as the width of NN increases and trivial with infinite width (the experiment results are in Section 5 and Table 2).

## 4.1 Finite-width Neural Network Trained by $\ell_2$ Regularized Loss

Inspired by [17], we can also show that every NN trained by (sub)gradient descent with loss function (7) is approximately a KM without the assumption of infinite width.

**Theorem 4.1.** *Suppose an NN $f(w, x)$, is learned from a training set $\{(x_i, y_i)\}_{i=1}^n$ by (sub)gradient descent with loss function (7) and gradient flow. Assume $sign(l'(y_i, f_t(x_i))) = sign(l'(y_i, f_0(x_i))), \forall t \in [0, T]$.[4] Then at some time $T > 0$,*

$$f_T(x) = \sum_{i=1}^n a_i K(x, x_i) + b, \quad with \quad K(x, x_i) = e^{-\lambda T} \int_0^T |l'(f_t(x_i), y_i)| \hat{\Theta}(w_t; x, x_i) e^{\lambda t} \, dt,$$

*and $a_i = -sign(l'(f_0(x_i), y_i)), b = e^{-\lambda T} f_0(x)$.*

See the proof in Appendix G, which utilizes the solution of inhomogeneous linear differential equation instead of integrating both side of dynamics (Eq. (9)) directly [17]. Note in Theorem 4.1, $a_i$ is deterministic and independent with $x$, different with [17] that has $a_i$ depends on $x$. Deterministic $a_i$ makes the function class simpler. Combing Theorem 4.1 with a bound of the Rademacher complexity of the KM [7] and a standard generalization using Rademacher complexity [37], we can compute the generalization bound of NN via the corresponding KM. See Appendix B for more background and experiments. The generalization bound we get will depend on $a_i$, which depends on the label $y_i$. This differs from traditional complexity measures that cannot explain the random label phenomenon [54].

## 4.2 Robustness of Infinite-width Neural Network

Our theory of equivalence allows us to deliver nontrivial robustness certificates for infinite-width NNs by considering the equivalent KMs. For an input $x_0 \in \mathbb{R}^d$, the objective of robustness is to find the largest ball such that no examples within this ball $x \in B(x_0, \delta)$ can change the classification result. Without loss of generality, we assume $g(x_0) > 0$. The robustness problem can be formulated as follows,

$$\max \delta, \quad \text{s.t. } g(x) > 0, \forall x \in B(x_0, \delta). \tag{10}$$

For an infinitely wide two-layer fully connected ReLU NN, $f(x) = \frac{1}{\sqrt{m}} \sum_{j=1}^m v_j \sigma(\frac{1}{\sqrt{d}} w_j^T x)$, where $\sigma(z) = \max(0, z)$ is the ReLU activation, the NTK is

$$\Theta(x, x') = \frac{\langle x, x' \rangle}{d} \left( \frac{\pi - \arccos(u)}{\pi} \right) + \frac{\|x\| \|x'\|}{2\pi d} \sqrt{1 - u^2}.$$

where $u = \frac{\langle x, x' \rangle}{\|x\| \|x'\|} \in [-1, 1]$. See the proof of the following theorem in Appendix H.1.

**Theorem 4.2.** *Consider the $\ell_\infty$ perturbation, for $x \in B_\infty(x_0, \delta) = \{x \in \mathbb{R}^d : \|x - x_0\|_\infty \leq \delta\}$, we can bound $\Theta(x, x')$ into some interval $[\Theta^L(x, x'), \Theta^U(x, x')]$. Suppose $g(x) = \sum_{i=1}^n \alpha_i \Theta(x, x_i)$, where $\alpha_i$ are known after solving the KM problems (e.g. SVM and KRR). Then we can lower bound $g(x)$ as follows.*

$$g(x) \geq \sum_{i=1, \alpha_i > 0}^n \alpha_i \Theta^L(x, x_i) + \sum_{i=1, \alpha_i < 0}^n \alpha_i \Theta^U(x, x_i).$$

Using a simple binary search and above theorem, we can find a lower bound for (10). Because of the equivalence between the infinite-width NN and KM, the lower bound we get for the KM is equivalently a robustness lower bound for the corresponding infinite-width NN.

---

[4]This is the case for hinge loss.

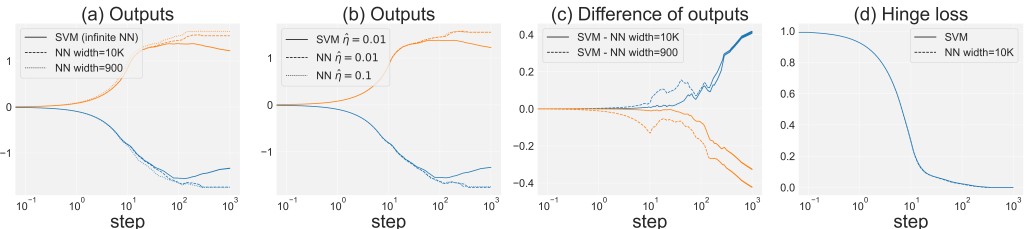

Figure 1: Training dynamics of neural network and SVM behave similarly. (a)(b) show dynamics of outputs for randomly selected two samples. (c) shows the difference between the outputs of SVM and NN. The dynamics of SVM agrees better with wider NN. (d) shows the dynamics of hinge loss for SVM and NN. Without specification, the width of NN is 10K and $\hat{\eta} = 0.1$.

## 5 Experiments

**(I) Verification of the equivalence.** The first experiment verifies the equivalence between soft margin SVM with NTK trained by subgradient descent and NN trained by soft margin loss. We train the SVM and 3-layer fully connected ReLU NN for a binary MNIST [28] classification (0 and 1) with learning rate $\hat{\eta} = 0.1$ and $\hat{\eta} = 0.01$ with full batch subgradient descent on $n = 128$ samples, where $\hat{\eta}$ is the learning rate used in experiments. Figure 1 shows the dynamics of the outputs and loss for NN and SVM. Since the regularization terms in the loss of NN and SVM are different, we just plot the hinge loss. It can be seen that the dynamics of NN and SVM agree very well. We also do a stochastic subgradient descent case for binary classification on full MNIST 0 and 1 data (12665 training and 2115 test) with learning rate $\hat{\eta} = 1$ and batch size 64, shown in Figure A.1. For more details, please see Appendix A.

**(II) Robustness of over-parameterized neural network.** Table 2 shows the robustness of two-layer overparameterized NNs with increasing width and SVM (which is equivalent to infinite-width two-layer ReLU NN) on binary classification of MNIST (0 and 1). We use the NN robustness verification algorithm (IBP) [22] to compute the robustness certificate for two-layer overparameterized NNs. The robustness certificate for SVM is computed using our method in Section 4.2. As demonstrated in Table 2, the certificate of NN almost decrease at a rate of $O(1/\sqrt{m})$ and will decrease to 0 as $m \to \infty$, where $m$ is the width of the hidden layer. We show that this is due to the bound propagation in Appendix H.2. Unfortunately, the decrease rate will be faster if the NN is deeper. The same problem will happen for LeCun initialization as well, which is used in PyTorch for fully connected layers by default. Notably, however, thanks to our theory, we could compute *nontrivial* robustness certificate for an infinite-width NN through the equivalent SVM as demonstrated.

Table 2: Robustness lower bounds of two-layer ReLU NN and SVM (infinite-width two-layer ReLU NN) tested on binary classification of MNIST (0 and 1). 100 test: randomly selected 100 test samples. Full test: full test data. Test only on data that classified correctly. std is computed over data samples. All models have test accuracy 99.95%. All values are mean of 5 experiments.

| Model | Width | Robustness certificate $\delta$ (mean $\pm$ std) $\times 10^{-3}$ | |
|---|---|---|---|
| | | 100 test | Full test |
| NN | $10^3$ | $7.4485 \pm 2.5667$ | $7.2708 \pm 2.1427$ |
| NN | $10^4$ | $2.9861 \pm 1.0730$ | $2.9367 \pm 0.89807$ |
| NN | $10^5$ | $0.99098 \pm 0.35775$ | $0.97410 \pm 0.29997$ |
| NN | $10^6$ | $0.31539 \pm 0.11380$ | $0.30997 \pm 0.095467$ |
| SVM | $\infty$ | $8.0541 \pm 2.5827$ | $7.9733 \pm 2.1396$ |

**(III) Comparison with kernel regression.** Table 3 compares our $\ell_2$ regularized models (KRR and SVM with NTK) with the previous kernel regression model ($\lambda = 0$ for KRR). All the robustness lower bounds are computed using our method in Section 4.2. While the accuracies of different models are similar, as the regularization increases, the robustness of KRR increases. The robustness of SVM outperforms the KRR with same regularization magnitude a lot. Our theory enables us to train an

equivalent infinite-width NN through SVM and KRR, which is intrinsically more robust than the previous kernel regression model.

Table 3: Robustness of equivalent infinite-width NN models with different loss functions (see Table 1) on binary classification of MNIST (0 and 1). $\lambda$ is the parameter in Eq. (7).

| | Model | $\lambda$ | Test accuracy | Robustness certificate $\delta$ | Robustness improvement |
|---|---|---|---|---|---|
| $\lambda = 0$([27, 4]) | KRR | 0 | 99.95% | $3.30202 \times 10^{-5}$ | - |
| | KRR | 0.001 | 99.95% | $3.756122 \times 10^{-5}$ | 1.14X |
| | KRR | 0.01 | 99.95% | $6.505500 \times 10^{-5}$ | 1.97X |
| | KRR | 0.1 | 99.95% | $2.229960 \times 10^{-4}$ | 6.75X |
| $\lambda > 0$ (ours) | KRR | 1 | 99.95% | 0.001005 | 30.43X |
| | KRR | 10 | 99.91% | 0.005181 | 156.90X |
| | KRR | 100 | 99.86% | 0.020456 | 619.50X |
| | KRR | 1000 | 99.76% | 0.026088 | 790.06X |
| | SVM | 0.064 | 99.95% | 0.008054 | 243.91X |

## 6  Conclusion and Future Works

In this paper, we establish the equivalence between SVM with NTK and the NN trained by soft margin loss with subgradient descent in the infinite width limit, and we show that they have the same dynamics and solution. We also extend our analysis to general $\ell_2$ regularized loss functions and show every neural network trained by such loss functions is approximately a KM. Finally, we demonstrate our theory is useful to compute *non-vacuous* generalization bound for NN, *non-trivial* robustness certificate for infinite-width NN while existing neural network robustness verification algorithm cannot handle, and with our theory, the resulting infinite-width NN from our $\ell_2$ regularized models is intrinsically more robust than that from the previous NTK kernel regression. For future research, since the equivalence between NN and SVM (and other $\ell_2$ regularized KMs) with NTK has been established, it would be very interesting to understand the generalization and robustness of NN from the perspective of these KMs. Our main results are currently still limited in the linear regime. It would be interesting to extend the results to the mean field setting or consider its connection with the implicit bias of NN.

## 7  Acknowledgement

We thank the anonymous reviewers for useful suggestions to improve the paper. We thank Libin Zhu for helpful discussions. We thank San Diego Supercomputer Center and MIT-IBM Watson AI Lab for computing resources. This work used the Extreme Science and Engineering Discovery Environment (XSEDE) [51], which is supported by National Science Foundation grant number ACI-1548562. This work used the Extreme Science and Engineering Discovery Environment (XSEDE) *Expanse* at San Diego Supercomputer Center through allocation TG-ASC150024 and ddp390. T.-W. Weng is partially supported by National Science Foundation under Grant No. 2107189.

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
