# Appendices

## A    Experiment Details

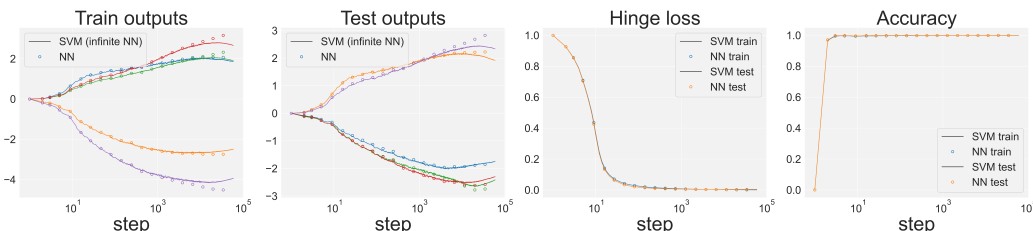

Figure A.1: SVM and NN trained by stochastic subgradient descent for binary MNIST classification task on full $0$ and $1$ data with learning rate $\hat{\eta} = 1$ and batch size $64$. The width of NN is $10$K.

### A.1    SVM Training

We use the following loss to train the SVM,

$$L(\beta) = \frac{\lambda}{2} \|\beta\|^2 + \frac{1}{n} \sum_{i=1}^{n} \max(0, 1 - y_i \langle \beta, \Phi(x_i) \rangle). \tag{11}$$

Let $\hat{\eta}$ be the learning rate for this loss in experiments. Then the dynamics of subgradient descent is

$$g_{t+1}(x) = (1 - \hat{\eta}\lambda)g_t(x) + \frac{\hat{\eta}}{n} \sum_{i=1}^{n} \mathbb{1}(y_i g_t(x_i) < 1) y_i K(x, x_i). \tag{12}$$

Denote $Q(t) = \frac{\hat{\eta}}{n} \sum_{i=1}^{n} \mathbb{1}(y_i g_t(x_i) < 1) y_i K(x, x_i)$, which is a linear combination of $K(x, x_i)$ and changes over time. The model output $g_t(x)$ at some time $T$ is

$$g_T(x) = (1 - \hat{\eta}\lambda)^T \left( g_0(x) + \frac{\hat{\eta}}{n} \sum_{t=0}^{T-1} (1 - \hat{\eta}\lambda)^{-t-1} Q(t) \right). \tag{13}$$

If we set $g_0(x) = 0$, we have

$$g_T(x) = \sum_{t=0}^{T-1} (1 - \hat{\eta}\lambda)^{T-1-t} Q(t). \tag{14}$$

We see that $g_T(x)$ is always a linear combination of kernel values $K(x, x_i)$ for $i = 1, \ldots, n$. Since $K(x, x_i)$ are fixed, we just need to store and update the weights of the kernel values. Let $\alpha_t \in \mathbb{R}^n$ be the weights at time $t$, that is

$$g_t(x) = \sum_{i=1}^{n} \alpha_{t,i} K(x, x_i). \tag{15}$$

Then according to Eq. (12), we update $\alpha$ at each subgradient descent step as follows.

$$\alpha_{t+1,i} = (1 - \hat{\eta}\lambda)\alpha_{t,i} + \frac{\hat{\eta}}{n} \mathbb{1}(y_i g_t(x_i) < 1) y_i, \quad \forall i \in \{1, \ldots, n\}. \tag{16}$$

For the SGD case, we sample $S_t \subseteq \{1, \ldots, n\}$ at step $t$ and update the weights of this subset while keep the other weights unchanged.

$$\alpha_{t+1,i} = (1 - \hat{\eta}\lambda)\alpha_{t,i} + \frac{\hat{\eta}}{|S_t|} \mathbb{1}(y_i g_t(x_i) < 1) y_i, \quad \forall i \in S_t,$$

$$\alpha_{t+1,i} = \alpha_{t,i}, \quad \forall i \notin S_t.$$

The kernelized implementation of Pegasos [44] set $\hat{\eta}_t = \frac{1}{\lambda t}$ for proving the convergence of the algorithm. In our experiments, we use constant $\hat{\eta}$.

## A.2 More Details

**(I) Verification of the equivalence.** The first experiment illustrates the equivalence between soft margin SVM with NTK trained by subgradient descent and NN trained by soft margin loss. We initialize 3-layer fully connected ReLU neural networks of width 10000 and 900, with NTK parameterization and make sure $f_0(x) = 0$ by subtracting the initial values from NN's outputs. We initialize the parameter of SVM with $\beta_0 = 0$, and this automatically makes sure $g_0(x) = 0$. SVM is trained by directly update the weights of kernel values [44] and more details can be found in Appendix A. We set the regularization parameter as $\lambda = 0.001$ and take the average of the hinge loss instead of sum.[5] We train the NN and SVM for a binary MNIST [28] classification task (0 and 1) with learning rate $\hat{\eta} = 0.1$ and $\hat{\eta} = 0.01$ with full batch subgradient descent on $n = 128$ samples, where $\hat{\eta}$ is the learning rate used in experiments (see Appendix A). Figure 1 shows the dynamics of the outputs and loss for NN and SVM. Since the regularization term in the loss of NN and SVM are different, we just plot the hinge loss. We see the dynamics of NN and SVM agree well. We also do a stochastic subgradient descent case for binary MNIST classification task on full 0 and 1 data (12665 train data and 2115 test data) with learning rate $\hat{\eta} = 1$ and batch size $64$, shown in Figure A.1.

Experiments are implemented with PyTorch [39] and the NTK of infinite-width NN is computed using Neural Tangents [38]. We do our experiments on 16G V100 GPU.

# B Computing Non-vacuous Generalization Bounds via Corresponding Kernel Machines

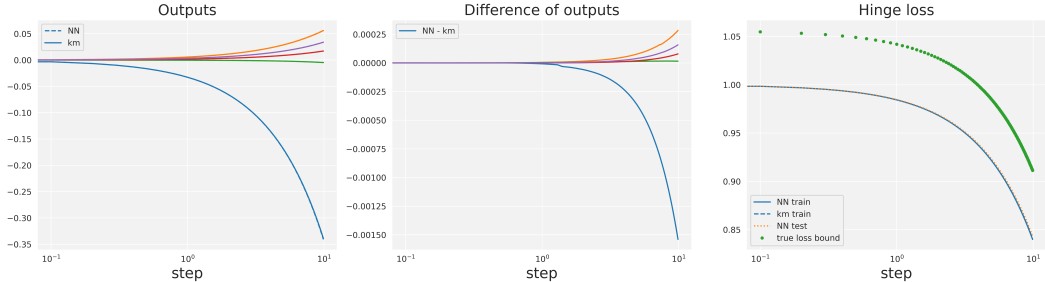

Figure B.2: Computing non-vacuous generalization bounds via corresponding kernel machines. Two-layer NN with 100 hidden nodes trained by full-batch subgradient descent for binary MNIST classification task on full 0 and 1 data with learning rate $\hat{\eta} = 0.1$. The kernel machine (KM) approximates NN very well. And we get a tight bound of the true loss by computing its Rademacher complexity. The confidence parameter is set as $1 - \delta = 0.99$.

Using Theorem 4.1, we can numerically compute the kernel machine that the NN is equivalent to, i.e. we can compute the kernel matrix and the weights at any time during the training. Then one can apply a generalization bound of kernel machines to give an generalization bound for this kernel machine (equivalently for this NN). Let $\mathcal{H}$ be the reproducing kernel Hilbert space (RKHS) corresponding to the kernel $K(\cdot, \cdot)$. The RKHS norm of a function $f(x) = \sum_{i=1}^n a_i K(x, x_i)$ is [6]

$$\|f\|_{\mathcal{H}} = \left\| \sum_{i=1}^n a_i \Phi(x_i) \right\| = \sqrt{\sum_{i=1}^n \sum_{j=1}^n a_i a_j K(x_i, x_j)}$$

**Lemma B.1** (Lemma 22 in [7]). *For a function class $\mathcal{F}_B = \{f(x) = \sum_{i=1}^n a_i K(x, x_i) : \|f\|_{\mathcal{H}} \leq B\} \subseteq \{x \to \langle \beta, \Phi(x) \rangle : \|\beta\| \leq B\}$, its empirical Rademacher complexity can be bounded as*

$$\hat{\mathcal{R}}_S(\mathcal{F}_B) = \frac{1}{n} \mathbb{E}_{\sigma_i \sim \{\pm 1\}^n} \left[ \sup_{f \in \mathcal{F}_B} \sum_{i=1}^n \sigma_i f(x_i) \right] \leq \frac{B}{n} \sqrt{\sum_{i=1}^n K(x_i, x_i)}$$

---

[5]This is equivalent to use $\lambda = 0.001 \times n$ in Eq. (7).

[6]Assume $f_0(x) = 0$.

Assume the data is sampled i.i.d. from some distribution $D$ and the population loss is $L_D(f) = \mathbb{E}_{(x,y) \sim D}[l(f(x), y)]$. The experical loss is $L_S(f) = \frac{1}{n}\sum_{i=1}^{n} l(f(x_i), y_i)$. Combing with a standard generalization bound using Rademacher complexity blow [37], we can get a bound of the population loss $L_D(f)$ for the kernel machine (equivalently for this NN).

**Lemma B.2.** *Suppose the loss $\ell(\cdot, \cdot)$ is bounded in $[0, c]$, and is $\rho$-Lipschitz in the first argument. Then with probability at least $1 - \delta$ over the sample $S$ of size $n$,*

$$\sup_{f \in \mathcal{F}}\{L_D(f) - L_S(f)\} \leq 2\rho\hat{\mathcal{R}}_S(\mathcal{F}) + 3c\sqrt{\frac{\log(2/\delta)}{2n}}$$

Most of the existing generalization bounds of NN [8, 35] are vacuous since they have a dependence on the number of parameters. Compared to those, the bound for kernel machines does not have a dependence on the number of NN's parameters, making it non-vacuous and promising. Moreover, we can even apply this generalization bound to optimize NN directly like PAC-Bayes bound [21], which gives NN with guaranteed generalization ability.

## C   Dynamics of Support Vector Machine

In this section, we derive the continuous and discrete dynamics of soft margin SVM trained by subgradient with the following loss function

$$L(\beta) = \frac{1}{2}\|\beta\|^2 + C\sum_{i=1}^{n}\max(0, 1 - y_i\langle\beta, \Phi(x_i)\rangle), \tag{17}$$

and the subgradient

$$\nabla_\beta L(\beta_t) = \beta_t - C\sum_{i=1}^{n}\mathbb{1}(y_i g_t(x_i) < 1)y_i\Phi(x_i). \tag{18}$$

**Lemma C.1.** *$L(\beta)$ satisfies the Polyak- Lojasiewicz (PL) inequality,*

$$L(\beta_t) - L(\beta^*) \leq \frac{1}{2}\|\nabla_\beta L(\beta_t)\|^2 \quad \forall\, \beta_t. \tag{19}$$

*where $\beta^* = \arg\min_\beta L(\beta)$.*

*Proof.* Since $L(\beta)$ is 1-strongly convex, by the definition of strong convexity and subgradient

$$L(\beta) \geq L(\beta_t) + \langle\nabla_\beta L(\beta_t), \beta - \beta_t\rangle + \frac{1}{2}\|\beta - \beta_t\|^2 \tag{20}$$

The right hand side is a convex quadratic function of $\beta$ (for fixed $\beta_t$). Setting the gradient with respect to $\beta$ equal to 0, we find that $\tilde{\beta} = \beta_t - \nabla_\beta L(\beta_t)$ minimize right hand side. Therefore we have

$$
\begin{aligned}
L(\beta) &\geq L(\beta_t) + \langle\nabla_\beta L(\beta_t), \beta - \beta_t\rangle + \frac{1}{2}\|\beta - \beta_t\|^2 \\
&\geq L(\beta_t) + \left\langle\nabla_\beta L(\beta_t), \tilde{\beta} - \beta_t\right\rangle + \frac{1}{2}\left\|\tilde{\beta} - \beta_t\right\|^2 \\
&= L(\beta_t) - \frac{1}{2}\|\nabla_\beta L(\beta_t)\|^2.
\end{aligned}
\tag{21}
$$

Since this holds for any $\beta$, we have

$$L(\beta^*) \geq L(\beta_t) - \frac{1}{2}\|\nabla_\beta L(\beta_t)\|^2. \tag{22}$$

$\square$

## C.1 Continuous Dynamics of SVM

Here we give the detailed derivation of the dynamics of soft margin SVM trained by subgradient. In the learning rate $\eta \to 0$ limit, the subgradient descent equation, which can also be written as

$$\frac{\beta_{t+1} - \beta_t}{\eta} = -\nabla_\beta L(\beta_t), \tag{23}$$

becomes a differential equation

$$\frac{d\beta_t}{dt} = -\nabla_\beta L(\beta_t). \tag{24}$$

This is known as gradient flow [2]. And we have defined the subgradient as

$$\nabla_\beta L(\beta_t) = \beta_t - C \sum_{i=1}^{n} \mathbb{1}(y_i g_t(x_i) < 1) y_i \Phi(x_i). \tag{25}$$

Applying the chain rule, the dynamics of $g_t(x) = \langle \beta_t, \Phi(x) \rangle$ is

$$\begin{aligned}
\frac{dg_t(x)}{dt} &= \frac{\partial g_t(x)}{\partial \beta_t} \frac{d\beta_t}{dt} \\
&= \left\langle \Phi(x), -\beta_t + C \sum_{i=1}^{n} \mathbb{1}(y_i g_t(x_i) < 1) y_i \Phi(x_i) \right\rangle \\
&= -g_t(x) + C \sum_{i=1}^{n} \mathbb{1}(y_i g_t(x_i) < 1) y_i K(x, x_i).
\end{aligned} \tag{26}$$

Denoting $Q(t) = C \sum_{i=1}^{n} \mathbb{1}(y_i g_t(x_i) < 1) y_i K(x, x_i)$, the equation becomes

$$\frac{dg_t(x)}{dt} + g_t(x) = Q(t). \tag{27}$$

Note this is a first-order inhomogeneous differential equation. The general solution at some time $T$ is given by

$$g_T(x) = e^{-T} \left( g_0(x) + \int_0^T Q(t) e^t \, dt \right). \tag{28}$$

As we already know that the loss function is strongly convex, $\beta$ will converge to the global optimizer in this infinite small learning rate setting. This can be seen by

$$\frac{d \left( L(\beta_t) - L(\beta^*) \right)}{dt} = \frac{dL(\beta_t)}{dt} = \frac{\partial L(\beta_t)}{\partial \beta_t} \frac{d\beta_t}{dt} = \langle \nabla_\beta L(\beta_t), -\nabla_\beta L(\beta_t) \rangle = - \|\nabla_\beta L(\beta_t)\|^2. \tag{29}$$

We see that $L(\beta_t)$ is always decreasing. Since $L(\beta)$ is strongly convex and thus bounded from below, by monotone convergence theorem, $L(\beta_t)$ will always converge. By Lemma C.1, we have the Polyak-Lojasiewicz (PL) inequality,

$$L(\beta_t) - L(\beta^*) \leq \frac{1}{2} \|\nabla_\beta L(\beta_t)\|^2 \tag{30}$$

Combining with above, we have

$$\frac{d \left( L(\beta_t) - L(\beta^*) \right)}{dt} \leq -2 \left( L(\beta_t) - L(\beta^*) \right). \tag{31}$$

Solving the equation, we get

$$L(\beta_t) - L(\beta^*) \leq e^{-2t} \left( L(\beta_0) - L(\beta^*) \right). \tag{32}$$

Thus we have a linear convergence rate.

Now, let us assume $g_T(x)$ will converge and see what is $g_T(x)$ as $T \to \infty$. As time increases $T \to \infty$, $e^{-T} g_0(x) \to 0$.

$$g_T(x) \to e^{-T} \int_0^T Q(t) e^t \, dt \tag{33}$$

$Q(t)$ is changing over time due to $g_t(x)$ is changing. Suppose $Q(t)$ keeps changing until some time $T_1$ and keeps constant, $Q(t) = Q$, after $T_1$,

$$\lim_{T \to \infty} g_T(x) = e^{-T} \int_0^{T_1} Q(t)e^t \, dt + e^{-T} \int_{T_1}^T Qe^t \, dt. \tag{34}$$

As $T \to \infty$, the first part of right hand side converges to 0.

$$\begin{aligned}
\lim_{T \to \infty} g_T(x) &\to e^{-T} \int_{T_1}^T Qe^t \, dt \\
&= e^{-T} \int_{T_1}^T e^t \, dt \cdot Q \\
&= e^{-T}(e^T - e^{T_1}) \cdot Q \\
&\to Q \\
&= C \sum_{i=1}^n \mathbb{1}(y_i g_T(x_i) < 1) \cdot y_i K(x, x_i).
\end{aligned} \tag{35}$$

## C.2 Discrete Dynamics of SVM

Let $\eta \in (0, 1)$ be the learning rate. The equation of subgradient descent update at some time $t$ is

$$\beta_{t+1} - \beta_t = -\eta \nabla_\beta L(\beta_t). \tag{36}$$

The dynamics of $g_t(x)$ is

$$\begin{aligned}
g_{t+1}(x) - g_t(x) &= \langle \beta_{t+1} - \beta_t, \Phi(x) \rangle \\
&= \left\langle -\eta \beta_t + \eta C \sum_{i=1}^n \mathbb{1}(y_i g_t(x_i) < 1) y_i \Phi(x_i), \Phi(x) \right\rangle \\
&= -\eta g_t(x) + \eta C \sum_{i=1}^n \mathbb{1}(y_i g_t(x_i) < 1) y_i K(x, x_i).
\end{aligned} \tag{37}$$

Denote second part as $Q(t) = \eta C \sum_{i=1}^n \mathbb{1}(y_i g_t(x_i) < 1) y_i K(x, x_i)$, which changes over time. The model $g_T(x)$ at some time $T$ is

$$\begin{aligned}
g_T(x) &= (1 - \eta)g_{T-1}(x) + Q(T - 1) \\
&= (1 - \eta)\left((1 - \eta)g_{T-2}(x) + Q(T - 2)\right) + Q(T - 1) \\
&= (1 - \eta)^T g_0(x) + \sum_{t=0}^{T-1}(1 - \eta)^{T-1-t}Q(t) \\
&= (1 - \eta)^T \left(g_0(x) + \sum_{t=0}^{T-1}(1 - \eta)^{-t-1}Q(t)\right).
\end{aligned} \tag{38}$$

The convergence of subgradient descent usually requires additional assumption that the norm of the subgradient is bounded. We refer readers to [44, 11, 9] for some proofs. Here let us assume the subgradient descent converges to the global optimizer and $Q(t)$ keeps changing until some time $T_1$

and keeps constant, $Q(t) = Q$, after $T_1$. As $T \to \infty$,

$$g_T(x) \to \sum_{t=0}^{T-1} (1 - \eta)^{T-1-t} Q(t)$$

$$= \sum_{t=0}^{T_1-1} (1 - \eta)^{T-1-t} Q(t) + \sum_{t=T_1}^{T-1} (1 - \eta)^{T-1-t} Q$$

$$\to \sum_{t=T_1}^{T-1} (1 - \eta)^{T-1-t} Q \tag{39}$$

$$= \sum_{t=T_1}^{T-1} (1 - \eta)^{T-1-t} Q$$

$$= \frac{-(1-\eta)^{T-T_1} + 1}{\eta} Q.$$

As $\eta \in (0, 1)$, $-(1-\eta)^{T-T_1} \to 0$.

$$g_T(x) \to \frac{1}{\eta} Q$$

$$= C \sum_{i=1}^{n} \mathbb{1}(y_i g_T(x_i) < 1) y_i K(x, x_i). \tag{40}$$

## D  Dynamics and Convergence Rate of Neural Network Trained by Soft Margin Loss

### D.1  Continuous Dynamics of NN

In the learning rate $\eta \to 0$ limit, the subgradient descent equation, which can also be written as

$$\frac{w_{t+1} - w_t}{\eta} = -\nabla_w L(w_t), \tag{41}$$

becomes a differential equation

$$\frac{dw_t}{dt} = -\nabla_w L(w_t). \tag{42}$$

This is known as gradient flow [2]. Then for any differentiable function $f_t(x)$,

$$\frac{df_t(x)}{dt} = \sum_{j=1}^{p} \frac{\partial f_t(x)}{\partial w_j} \frac{dw_j}{dt}, \tag{43}$$

where $p$ is the number of parameters. Replacing $\frac{dw_j}{dt}$ by its subgradient descent expression:

$$\frac{df_t(x)}{dt} = \sum_{j=1}^{p} \frac{\partial f_t(x)}{\partial w_j} \left( -\frac{\partial L(w_t)}{\partial w_j} \right). \tag{44}$$

And we know

$$\frac{\partial L(w_t)}{\partial w_j} = w_j \mathbb{1}(w_j \in W^{(L+1)}) + \sum_{i=1}^{n} \frac{\partial L_h}{\partial f_t(x_i)} \frac{\partial f_t(x_i)}{\partial w_j}. \tag{45}$$

where $\mathbb{1}(w_j \in W^{(L+1)})$ equals to 1 if the parameter $w_j$ is in the last layer $W^{(L+1)}$ else 0. Combining above together,

$$\frac{df_t(x)}{dt} = \sum_{j=1}^{p} \frac{\partial f_t(x)}{\partial w_j} \left( -w_j \mathbb{1}(w_j \in W^{(L+1)}) - \sum_{i=1}^{n} \frac{\partial L_h}{\partial f_t(x_i)} \frac{\partial f_t(x_i)}{\partial w_j} \right). \tag{46}$$

Rearranging terms:

$$\frac{df_t(x)}{dt} = -\sum_{k=1}^{p_{L+1}} \frac{\partial f_t(x)}{\partial W_k^{(L+1)}} W_k^{(L+1)} - \sum_{i=1}^{n} \frac{\partial L_h}{\partial f_t(x_i)} \sum_{j=1}^{p} \frac{\partial f_t(x)}{\partial w_j} \frac{\partial f_t(x_i)}{\partial w_j}, \tag{47}$$

where $p_{L+1}$ is the number of parameters of the last layer ($L + 1$ layer). The first part of the right hand side is

$$\sum_{k=1}^{p_{L+1}} \frac{\partial f_t(x)}{\partial W_k^{(L+1)}} W_k^{(L+1)} = \left\langle \frac{\partial f_t(x)}{\partial W^{(L+1)}}, W^{(L+1)} \right\rangle = \left\langle \alpha_t^{(L)}(x), W^{(L+1)} \right\rangle = f_t(x). \tag{48}$$

Applying $L_h'(y_i, f_t(x_i)) = \frac{\partial L_h}{\partial f_t(x_i)}$, the subgradient of hinge loss, and the definition of tangent kernel (2.1), the second part is

$$-\sum_{i=1}^{n} \frac{\partial L_h}{\partial f_t(x_i)} \sum_{j=1}^{p} \frac{\partial f_t(x)}{\partial w_j} \frac{\partial f_t(x_i)}{\partial w_j} = -\sum_{i=1}^{n} L_h'(y_i, f_t(x_i)) \hat{\Theta}(w_t; x, x_i). \tag{49}$$

Thus the equation becomes

$$\frac{df_t(x)}{dt} = -f_t(x) - \sum_{i=1}^{n} L_h'(y_i, f_t(x_i)) \hat{\Theta}(w_t; x, x_i). \tag{50}$$

Take $L_h'(y_i, f_t(x_i)) = -C y_i \mathbb{1}(y_i f_t(x_i) < 1)$ in

$$\frac{df_t(x)}{dt} = -f_t(x) + C \sum_{i=1}^{n} \mathbb{1}(y_i f_t(x_i) < 1) y_i \hat{\Theta}(w_t; x, x_i). \tag{51}$$

### D.2 Additional Notations

Denote $X \in \mathbb{R}^{d \times n}$ as the training data. Denote $f_t = f_t(X) \in \mathbb{R}^n$ and $g_t = g_t(X) \in \mathbb{R}^n$ as the outputs of NN and SVM on the training data. Denote $\hat{\Theta}(w_t) = \hat{\Theta}(w_t; X, X) \in \mathbb{R}^{n \times n}$ as the tangent kernel evaluated on the training data at time $t$, and $l'(f_t) \in \mathbb{R}^n$ as the derivative of the loss function w.r.t. $f_t$. Denote $\nabla_w f_t \in \mathbb{R}^{n \times p}$ as the Jacobian and we have $\hat{\Theta}(w_t) = \nabla_w f_t \nabla_w f_t^T$. Denote $\lambda_0 = \lambda_{min}\left(\hat{\Theta}(w_t)\right)$ as the smallest eigenvalue of $\hat{\Theta}(w_t)$. Then we can write the dynamics of NN as

$$\frac{d}{dt} f_t = -f_t - \hat{\Theta}(w_t) l'(f_t).$$

Let $v \in \mathbb{R}^p$ with $v_j = \mathbb{1}(w_j \in W^{(L+1)})$. We can write the gradient as

$$\nabla_w L(w_t) = w_t \odot v + \nabla_w f_t^T l'(f_t).$$

### D.3 Convergence of NN

The loss of NN is

$$L(w_t) = \frac{1}{2} \left\| W_t^{(L+1)} \right\|^2 + \sum_i^n l(f_t(x_i), y_i),$$

where $l(f, y) = C \max(0, 1 - yf)$. The dynamic of the loss is

$$\frac{dL(w_t)}{dt} = \frac{\partial L(w_t)}{\partial w_t} \frac{dw_t}{dt} = \langle \nabla_w L(w_t), -\nabla_w L(w_t) \rangle = -\|\nabla_w L(w_t)\|^2.$$

Since $L(w_t) \geq 0$ is bounded from below, by monotone convergence theorem, $L(w_t)$ will always converge to a stationary point. Applying Lemma D.1, we have

$$\frac{d\left(L(w_t) - L(w^*)\right)}{dt} = -\|\nabla_w L(w_t)\|^2 \leq -2\left(L(w_t) - L(w^*)\right).$$

Thus we have a linear convergence, same as SVM.

$$L(w_t) - L(w^*) \leq e^{-2t}\left(L(w_0) - L(w^*)\right).$$

**Lemma D.1** (PL inequality of NN for soft margin loss). *Assume $\lambda_0 \geq \frac{2}{C}$, then $L(w_t)$ satisfies the PL condition*

$$\|\nabla_w L(w_t)\|^2 \geq 2\left(L(w_t) - L(w^*)\right).$$

*Proof.*

$$
\begin{aligned}
\|\nabla_w L(w_t)\|^2 &= \left\langle w_t \odot v + \nabla_w f_t^T l'(f_t), w_t \odot v + \nabla_w f_t^T l'(f_t) \right\rangle \\
&= \left\langle w_t \odot v, w_t \odot v \right\rangle + \left\langle \nabla_w f_t^T l'(f_t), \nabla_w f_t^T l'(f_t) \right\rangle + 2 \left\langle w_t \odot v, \nabla_w f_t^T l'(f_t) \right\rangle \\
&= \left\| W_t^{(L+1)} \right\|^2 + l'(f_t)^T \hat{\Theta}(w_t) l'(f_t) + 2 \left\langle W_t^{(L+1)}, \nabla_{W^{(L+1)}} f_t^T l'(f_t) \right\rangle \\
&= \left\| W_t^{(L+1)} \right\|^2 + l'(f_t)^T \hat{\Theta}(w_t) l'(f_t) + 2 f_t^T l'(f_t).
\end{aligned}
$$

We want the loss satisfies the PL condition $\|\nabla_w L(w_t)\|^2 \geq 2\left(L(w_t) - L(w^*)\right)$.

$$
\begin{aligned}
&\|\nabla_w L(w_t)\|^2 - 2\left(L(w_t) - L(w^*)\right) \\
&= \|\nabla_w L(w_t)\|^2 - 2L(w_t) + 2L(w^*) \\
&= l'(f_t)^T \hat{\Theta}(w_t) l'(f_t) + 2 f_t^T l'(f_t) - 2 \sum_i^n l(f_t(x_i), y_i) + 2L(w^*) \\
&\geq \lambda_0 \|l'(f_t)\|^2 + 2 f_t^T l'(f_t) - 2 \sum_i^n l(f_t(x_i), y_i) + 2L(w^*),
\end{aligned}
$$

where the last inequality is the inequality of quadratic form. For hinge loss $l(f, y) = C\max(0, 1 - yf) = C(1 - yf)\mathbb{1}(1 - yf > 0)$ and $l'(f, y) = -Cy\mathbb{1}(1 - yf > 0)$,

$$
\begin{aligned}
&\|\nabla_w L(w_t)\|^2 - 2\left(L(w_t) - L(w^*)\right) \\
&\geq \lambda_0 \|l'(f_t)\|^2 + 2 f_t^T l'(f_t) - 2 \sum_i^n l(f_t(x_i), y_i) + 2L(w^*) \\
&= \lambda_0 \sum_i^n l'(f_t(x_i), y_i)^2 + 2 \sum_i^n f_t(x_i) l'(f_t(x_i), y_i) - 2 \sum_i^n l(f_t(x_i), y_i) + 2L(w^*) \\
&= \lambda_0 \sum_i^n C^2 \mathbb{1}(1 - y_i f_t(x_i) > 0) - 2 \sum_i^n C y_i f_t(x_i) \mathbb{1}(1 - y_i f_t(x_i) > 0) \\
&\quad - 2 \sum_i^n C(1 - y_i f_t(x_i)) \mathbb{1}(1 - y_i f_t(x_i) > 0) + 2L(w^*) \\
&= C \sum_i^n \mathbb{1}(1 - y_i f_t(x_i) > 0) \left(C\lambda_0 - 2\right) + 2L(w^*).
\end{aligned}
$$

Since $L(w^*) > 0$, as long as $\lambda_0 \geq 2/C$, the loss $L(w_t)$ satisfies the PL condition $\|\nabla_w L(w_t)\|^2 \geq 2\left(L(w_t) - L(w^*)\right)$. $\lambda_0 \geq 2/C$ can be guaranteed in a parameter ball when $\frac{2}{C} < \lambda_{min}\left(\hat{\Theta}(w_0)\right)$ by using a sufficiently wide NN [33]. $\square$

### D.4 Discrete Dynamics of NN

The subgradient descent update is

$$w_{t+1} - w_t = -\eta \nabla_w L(w_t). \tag{52}$$

We consider the situation of constant NTK, $\hat{\Theta}(w_t; x, x_i) \to \hat{\Theta}(w_0; x, x_i)$, or equivalently linear model. As proved by Proposition 2.2 in [32], the tangent kernel of a differentiable function $f(w, x)$

is constant if and only if $f(w, x)$ is linear in $w$. Take the Taylor expansion of $f(w_{t+1}, x)$ at $w_t$,

$$
\begin{aligned}
& f(w_{t+1}, x) - f(w_t, x) \\
&= f(w_t, x) + \langle \nabla_w f(w_t, x), w_{t+1} - w_t \rangle - f(w_t, x) \\
&= \langle \nabla_w f(w_t, x), -\eta \nabla_w L(w_t) \rangle \\
&= \left\langle \nabla_w f(w_t, x), -\eta \left( wv + \sum_{i=1}^{n} L'_h(y_i, f_t(x_i)) \nabla_w f_t(x_i) \right) \right\rangle \\
&= -\eta f_t(x) + \eta \sum_{i=1}^{n} L'_h(y_i, f_t(x_i)) \hat{\Theta}(w_t; x, x_i) \\
&= -\eta f_t(x) + \eta C \sum_{i=1}^{n} \mathbb{1}(y_i f_t(x_i) < 1) y_i \hat{\Theta}(w_t; x, x_i) \\
& \to -\eta f_t(x) + \eta C \sum_{i=1}^{n} \mathbb{1}(y_i f_t(x_i) < 1) y_i \hat{\Theta}(w_0; x, x_i).
\end{aligned}
\tag{53}
$$

# E    Proof of Theorem 3.4

*Proof.* We prove the constancy of tangent kernel by adopting the results of [34].

**Lemma E.1** (Theorem 3.3 in [32]; Hessian norm is controlled by the minimum hidden layer width)**.** *Consider a general neural network $f(w, x)$ of the form Eq. (1), which can be a fully connected network, CNN, ResNet or a mixture of these types. Let $m$ be the minimum of the hidden layer widths, i.e., $m = \min_{l \in [L]} m_l$. Given any fixed $R > 0$, and any $w \in B(w_0; R) := \{w : \|w - w_0\| \leq R\}$, with high probability over the initialization, the Hessian spectral norm satisfies the following:*

$$
\|H(w)\| = O\left(\frac{R^{3L} \ln m}{\sqrt{m}}\right).
\tag{54}
$$

**Lemma E.2** (Proposition 2.3 in [32]; Small Hessian norm $\Rightarrow$ Small change of tangent kernel)**.** *Given a point $w_0 \in \mathbb{R}^p$ and a ball $B(w_0; R) := \{w : \|w - w_0\| \leq R\}$ with fixed radius $R > 0$, if the Hessian matrix satisfies $\|H(w)\| < \epsilon$, where $\epsilon > 0$, for all $w \in B(w_0, R)$, then the tangent kernel $\hat{\Theta}(w; x, x')$ of the model, as a function of $w$, satisfies*

$$
\left| \hat{\Theta}(w; x, x') - \hat{\Theta}(w_0; x, x') \right| = O(\epsilon R), \quad \forall w \in B(w_0; R), \ \forall x, x' \in \mathbb{R}^d.
\tag{55}
$$

Applying above two lemmas, we can see that in the limit of $m \to \infty$, the spectral norm of Hessian converge to 0 and the tangent kernel keeps constant in the ball $B(w_0; R)$.

**Corollary E.2.1** (Consistancy of tangent kernel)**.** *Consider a general neural network $f(w, x)$ of the form Eq. (1). Given a point $w_0 \in \mathbb{R}^p$ and a ball $B(w_0; R) := \{w : \|w - w_0\| \leq R\}$ with fixed radius $R > 0$, in the infinite width limit, $m \to \infty$,*

$$
\lim_{m \to \infty} \hat{\Theta}(w; x, x') \to \hat{\Theta}(w_0; x, x_i), \quad \forall w \in B(w_0; R), \ \forall x, x' \in \mathbb{R}^d.
\tag{56}
$$

Thus we prove the constancy of tangent kernel in infinite width limit. Then it is easy to check the dynamics of infinitely wide NN is the same with the dynamics of SVM with constant NTK.

$\square$

# F    Bound the difference between SVM and NN

Assume the loss $l$ is $\rho$-lipschitz and $\beta_l$-smooth for the first argument (i.e. the model output). Assume $f_0(x) = g_0(x)$ for any $x$.

## F.1 Bound the difference on the Training Data

The dynamics of the NN and SVM are

$$\frac{d}{dt}f_t = -\lambda f_t - \hat{\Theta}(w_t)l'(f_t)$$

$$\frac{d}{dt}g_t = -\lambda g_t - \hat{\Theta}(w_0)l'(g_t)$$

The dynamics of the difference between them is

$$\frac{d}{dt}(f_t - g_t) = -\lambda(f_t - g_t) - \left(\hat{\Theta}(w_t)l'(f_t) - \hat{\Theta}(w_0)l'(g_t)\right)$$

The solution of the above differential equation at time $T$ is

$$f_T - g_T = e^{-\lambda T}(f_0 - g_0) - e^{-\lambda T}\int_0^T \left(\hat{\Theta}(w_t)l'(f_t) - \hat{\Theta}(w_0)l'(g_t)\right)e^{\lambda t}dt$$

$$= e^{-\lambda T}\int_0^T \left(\hat{\Theta}(w_0)l'(g_t) - \hat{\Theta}(w_t)l'(f_t)\right)e^{\lambda t}dt$$

using $f_0 = g_0$. Thus

$$\|f_T - g_T\| \le e^{-\lambda T}\int_0^T \left\|\hat{\Theta}(w_0)l'(g_t) - \hat{\Theta}(w_t)l'(f_t)\right\|e^{\lambda t}dt$$

Since $l$ is $\beta_l$ smooth,

$$\left\|\hat{\Theta}(w_0)l'(g_t) - \hat{\Theta}(w_t)l'(f_t)\right\| = \left\|\hat{\Theta}(w_0)l'(g_t) - \hat{\Theta}(w_0)l'(f_t) + \hat{\Theta}(w_0)l'(f_t) - \hat{\Theta}(w_t)l'(f_t)\right\|$$

$$= \left\|\hat{\Theta}(w_0)\left(l'(g_t) - l'(f_t)\right) + \left(\hat{\Theta}(w_0) - \hat{\Theta}(w_t)\right)l'(f_t)\right\|$$

$$\le \left\|\hat{\Theta}(w_0)\left(l'(g_t) - l'(f_t)\right)\right\| + \left\|\left(\hat{\Theta}(w_0) - \hat{\Theta}(w_t)\right)l'(f_t)\right\|$$

$$\le \beta_l\left\|\hat{\Theta}(w_0)\right\|\|g_t - f_t\| + \rho\sqrt{n}\left\|\hat{\Theta}(w_0) - \hat{\Theta}(w_t)\right\|$$

where $\|l'(f_t)\| \le \rho\sqrt{n}$. Thus we have

$$\|f_T - g_T\| \le e^{-\lambda T}\beta_l\left\|\hat{\Theta}(w_0)\right\|\int_0^T \|g_t - f_t\|e^{\lambda t}dt + e^{-\lambda T}\rho\sqrt{n}\int_0^T \left\|\hat{\Theta}(w_0) - \hat{\Theta}(w_t)\right\|e^{\lambda t}dt$$

Applying the Grönwall's inequality,

$$\|f_T - g_T\| \le e^{-\lambda T}\rho\sqrt{n}\int_0^T \left\|\hat{\Theta}(w_0) - \hat{\Theta}(w_t)\right\|e^{\lambda t}dt \cdot e^{e^{-\lambda T}\beta_l\|\hat{\Theta}(w_0)\|\int_0^T e^{\lambda t}dt}$$

$$= e^{-\lambda T}\rho\sqrt{n}\int_0^T \left\|\hat{\Theta}(w_0) - \hat{\Theta}(w_t)\right\|e^{\lambda t}dt \cdot e^{\frac{1}{\lambda}(1-e^{-\lambda T})\beta_l\|\hat{\Theta}(w_0)\|}$$

$$= e^{-\lambda T}e^{\frac{1}{\lambda}(1-e^{-\lambda T})\beta_l\|\hat{\Theta}(w_0)\|}\rho\sqrt{n}\int_0^T \left\|\hat{\Theta}(w_0) - \hat{\Theta}(w_t)\right\|e^{\lambda t}dt$$

By Lemma E.1 and Lemma E.2, in a parameter ball $B(w_0; R) = \{w : \|w - w_0\| \le R\}$, with high probability, $\left|\hat{\Theta}(w; x, x') - \hat{\Theta}(w_0; x, x')\right| = O(R^{3L+1}\ln m/\sqrt{m})$ w.r.t. $m$. Then we have

$$\left\|\hat{\Theta}(w_0) - \hat{\Theta}(w_t)\right\| \le \left\|\hat{\Theta}(w_0) - \hat{\Theta}(w_t)\right\|_F = O(\frac{R^{3L+1}n\ln m}{\sqrt{m}})$$

Thus we have

$$\|f_T - g_T\| \le \frac{1}{\lambda}(1 - e^{-\lambda T})e^{(1-e^{-T})\beta_l\|\hat{\Theta}(w_0)\|}\rho\sqrt{n} \cdot O(\frac{R^{3L+1}n\ln m}{\sqrt{m}})$$

$$= O(\frac{e^{\beta_l\|\hat{\Theta}(w_0)\|}R^{3L+1}\rho n^{\frac{3}{2}}\ln m}{\lambda\sqrt{m}})$$

## F.2 Bound on the Test Data

For a test data $x$, the prove is similar to the training case. Denote $\hat{\Theta}(w_t; X, x) \in \mathbb{R}^n$ as the tangent kernel evaluate between the training data and a test data $x$. Recall

$$\frac{df_t(x)}{dt} = -\lambda f_t(x) - \hat{\Theta}(w_t; X, x)^T l'(f_t)$$

$$\frac{dg_t(x)}{dt} = -\lambda g_t(x) - \hat{\Theta}(w_0; X, x)^T l'(g_t)$$

$$\frac{d}{dt}\left(f_t(x) - g_t(x)\right) = -\lambda\left(f_t(x) - g_t(x)\right) - \left(\hat{\Theta}(w_t; X, x)^T l'(f_t) - \hat{\Theta}(w_0; X, x)^T l'(g_t)\right)$$

The solution of the above differential equation is

$$f_T(x) - g_T(x) = e^{-\lambda T}\left(f_0 - g_0\right) - e^{-\lambda T}\int_0^T \left(\hat{\Theta}(w_t; X, x)^T l'(f_t) - \hat{\Theta}(w_0; X, x)^T l'(g_t)\right)e^{\lambda t}dt$$

$$= e^{-\lambda T}\int_0^T \left(\hat{\Theta}(w_0; X, x)^T l'(g_t) - \hat{\Theta}(w_t; X, x)^T l'(f_t)\right)e^{\lambda t}dt$$

using $f_0 = g_0$. Thus

$$\|f_T(x) - g_T(x)\| \le e^{-\lambda T}\int_0^T \left\|\hat{\Theta}(w_0; X, x)^T l'(g_t) - \hat{\Theta}(w_t; X, x)^T l'(f_t)\right\|e^{\lambda t}dt$$

Since $l$ is $\beta_l$ smooth,

$$\left\|\hat{\Theta}(w_0; X, x)^T l'(g_t) - \hat{\Theta}(w_t; X, x)^T l'(f_t)\right\|$$

$$= \left\|\hat{\Theta}(w_0; X, x)^T l'(g_t) - \hat{\Theta}(w_0; X, x)^T l'(f_t) + \hat{\Theta}(w_0; X, x)^T l'(f_t) - \hat{\Theta}(w_t; X, x)^T l'(f_t)\right\|$$

$$= \left\|\hat{\Theta}(w_0; X, x)^T \left(l'(g_t) - l'(f_t)\right) + \left(\hat{\Theta}(w_0; X, x)^T - \hat{\Theta}(w_t; X, x)^T\right)l'(f_t)\right\|$$

$$\le \left\|\hat{\Theta}(w_0; X, x)^T \left(l'(g_t) - l'(f_t)\right)\right\| + \left\|\left(\hat{\Theta}(w_0; X, x)^T - \hat{\Theta}(w_t; X, x)^T\right)l'(f_t)\right\|$$

$$\le \beta_l \left\|\hat{\Theta}(w_0; X, x)\right\|\|g_t - f_t\| + \rho\sqrt{n}\left\|\left(\hat{\Theta}(w_0; X, x)^T - \hat{\Theta}(w_t; X, x)^T\right)\right\|$$

where $\|l'(f_t)\| \le \rho\sqrt{n}$. Thus we have

$$\|f_T(x) - g_T(x)\|$$

$$\le e^{-\lambda T}\beta_l\left\|\hat{\Theta}(w_0; X, x)\right\|\int_0^T \|g_t - f_t\|e^{\lambda t}dt + e^{-\lambda T}\rho\sqrt{n}\int_0^T \left\|\hat{\Theta}(w_0; X, x)^T - \hat{\Theta}(w_t; X, x)^T\right\|e^{\lambda t}dt$$

Applying the Grönwall's inequality,

$$\|f_T(x) - g_T(x)\|$$

$$\le e^{-\lambda T}\rho\sqrt{n}\int_0^T \left\|\hat{\Theta}(w_0; X, x)^T - \hat{\Theta}(w_t; X, x)^T\right\|e^{\lambda t}dt \cdot e^{e^{-\lambda T}\beta_l\|\hat{\Theta}(w_0;X,x)\|\int_0^T e^{\lambda t}dt}$$

$$= e^{-\lambda T}\rho\sqrt{n}\int_0^T \left\|\hat{\Theta}(w_0; X, x)^T - \hat{\Theta}(w_t; X, x)^T\right\|e^{\lambda t}dt \cdot e^{\frac{1}{\lambda}(1-e^{-\lambda T})\beta_l\|\hat{\Theta}(w_0;X,x)\|}$$

$$= e^{-\lambda T}e^{\frac{1}{\lambda}(1-e^{-\lambda T})\beta_l\|\hat{\Theta}(w_0;X,x)\|}\rho\sqrt{n}\int_0^T \left\|\hat{\Theta}(w_0; X, x)^T - \hat{\Theta}(w_t; X, x)^T\right\|e^{\lambda t}dt$$

By Lemma E.1 and Lemma E.2, in a parameter ball $B(w_0; R) = \{w : \|w - w_0\| \le R\}$, with high probability, $\left|\hat{\Theta}(w; x, x') - \hat{\Theta}(w_0; x, x')\right| = O(R^{3L+1}\ln m/\sqrt{m})$. Then we have

$$\left\|\hat{\Theta}(w_0; X, x)^T - \hat{\Theta}(w_t; X, x)^T\right\| = O(\frac{R^{3L+1}\sqrt{n}\ln m}{\sqrt{m}})$$

Thus we have

$$\|f_T(x) - g_T(x)\| \le \frac{1}{\lambda}(1 - e^{-\lambda T})e^{(1-e^{-T})\beta_l}\|\hat{\Theta}(w_0; X, x)\|\rho\sqrt{n} \cdot O(\frac{R^{3L+1}\sqrt{n}\ln m}{\sqrt{m}})$$

$$= O(\frac{e^{\beta_l\|\hat{\Theta}(w_0; X, x)\|}R^{3L+1}\rho n \ln m}{\lambda\sqrt{m}})$$

# G  Finite-width Neural Networks are Kernel Machines

Inspired by [17], we can also show that every neural network trained by (sub)gradient descent with loss function in the form (7) is approximately a kernel machine without the assumption of infinite width limit.

**Theorem G.1.** *Suppose a neural network $f(w, x)$, with $f$ a differentiable function of $w$, is learned from a training set $\{(x_i, y_i)\}_{i=1}^n$ by (sub)gradient descent with loss function $L(w) = \frac{\lambda}{2}\|W^{(L+1)}\|^2 + \sum_{i=1}^n l(y_i, f(w, x_i))$ and gradient flow. Assume $sign(l'(y_i, f_t(x_i))) = sign(l'(y_i, f_0(x_i))), \forall t \in [0, T]$, keeps unchanged during training. Then at some time $T$,*

$$f_T(x) = \sum_{i=1}^n a_i K(x, x_i) + b, \tag{57}$$

*where*

$$a_i = -sign(l'(y_i, f_0(x_i))), \qquad b = e^{-\lambda T}f_0(x),$$

$$K(x, x_i) = e^{-\lambda T}\int_0^T |l'(y_i, f_t(x_i))|\,\hat{\Theta}(w_t; x, x_i)e^{\lambda t}\,dt\,dt$$

*Proof.* As we have derived, the neural network follows the dynamics of Eq. (9):

$$\frac{df_t(x)}{dt} = -\lambda f_t(x) - \sum_{i=1}^n l'(y_i, f_t(x_i))\hat{\Theta}(w_t; x, x_i). \tag{58}$$

Note this is a first-order inhomogeneous linear differential equation with the functions depended on $t$. Denote $Q(t) = -\sum_{i=1}^n l'(y_i, f_t(x_i))\hat{\Theta}(w_t; x, x_i)$,

$$\frac{df_t(x)}{dt} + \lambda f_t(x) = Q(t). \tag{59}$$

Let $f_0(x)$ be the initial model, prior to gradient descent. The solution is given by

$$f_T(x) = e^{-\lambda T}\left(f_0(x) + \int_0^T Q(t)e^{\lambda t}\,dt\right). \tag{60}$$

Then

$$f_T(x) = e^{-\lambda T}\left(f_0(x) - \sum_{i=1}^n \int_0^T l'(y_i, f_t(x_i))\hat{\Theta}(w_t; x, x_i)e^{\lambda t}\,dt\right)$$

$$= e^{-\lambda T}f_0(x) - \sum_{i=1}^n e^{-\lambda T}\int_0^T l'(y_i, f_t(x_i))\hat{\Theta}(w_t; x, x_i)e^{\lambda t}\,dt$$

$$= e^{-\lambda T}f_0(x) - \sum_{i=1}^n e^{-\lambda T}\int_0^T sign(l'(y_i, f_t(x_i))) \cdot |l'(y_i, f_t(x_i))|\,\hat{\Theta}(w_t; x, x_i)e^{\lambda t}\,dt$$

$$= e^{-\lambda T}f_0(x) - \sum_{i=1}^n sign(l'(y_i, f_0(x_i))) \cdot e^{-\lambda T}\int_0^T |l'(y_i, f_t(x_i))|\,\hat{\Theta}(w_t; x, x_i)e^{\lambda t}\,dt.$$

$$\tag{61}$$

where the last equality uses the assumption $\text{sign}(l'(y_i, f_t(x_i))) = \text{sign}(l'(y_i, f_0(x_i))), \forall t \in [0, T]$. Thus

$$f_T(x) = \sum_{i=1}^{n} a_i K(x, x_i) + b, \tag{62}$$

with

$$a_i = -\text{sign}(l'(y_i, f_0(x_i))), \qquad b = e^{-\lambda T} f_0(x),$$

$$K(x, x_i) = e^{-\lambda T} \int_0^T |l'(y_i, f_t(x_i))| \, \hat{\Theta}(w_t; x, x_i) e^{\lambda t} \, dt$$

$\square$

$K(x, x_i) = e^{-\lambda T} \int_0^T |l'(y_i, f_t(x_i))| \, \hat{\Theta}(w_t; x, x_i) e^{\lambda t} \, dt$ is a valid kernel since it is a nonnegative sum of positive definite kernels. Our $a_i$, $b$ and $K(x, x_i)$ will stay bounded as long as $f_0(x)$, $l'(y_i, f_t(x_i))$ and $\hat{\Theta}(w_t; x, x_i)$ are bounded.

## H  Robustness of Over-parameterized Neural Network

### H.1  Robustness Verification of NTK

For an infinitely wide two-layer fully connected ReLU NN, $f(x) = \frac{1}{\sqrt{m}} \sum_{j=1}^{m} v_j \sigma(\frac{1}{\sqrt{d}} w_j^T x)$, where $\sigma(z) = \max(0, z)$ is the ReLU activation. The NTK is

$$\Theta(x, x') = \frac{\langle x, x' \rangle}{d} \left( \frac{\pi - \arccos(u)}{\pi} \right) + \frac{\|x\| \, \|x'\|}{2\pi d} \sqrt{1 - u^2} = \frac{\|x\| \, \|x'\|}{2\pi d} h(u), \tag{63}$$

$$h(u) = 2u(\pi - \arccos(u)) + \sqrt{1 - u^2}. \tag{64}$$

where $u = \frac{\langle x, x' \rangle}{\|x\| \|x'\|} \in [-1, 1]$. Consider the $\ell_\infty$ perturbation, for $x \in B_\infty(x_0, \delta) = \{x \in \mathbb{R}^d : \|x - x_0\|_\infty \leq \delta\}$, we can bound $\|x\|$ in the interval $[\|x\|^L, \|x\|^U]$ as follows.

$$\|x\| = \|x_0 + \Delta\| \leq \|x_0\| + \|\Delta\| \leq \|x_0\| + \sqrt{d}\delta = \|x\|^U,$$

$$\|x\| = \|x_0 + \Delta\| \geq \|\|x_0\| - \|\Delta\|\| \geq \max(\|x_0\| - \sqrt{d}\delta, 0) = \|x\|^L.$$

Then we can also bound $u$ in $[u^L, u^U]$.

$$\langle x, x' \rangle = \langle x_0 + \Delta, x' \rangle \in \left[ \langle x_0, x' \rangle - \sqrt{d}\delta \|x'\|, \langle x_0, x' \rangle + \sqrt{d}\delta \|x'\| \right],$$

$$u^L = \frac{\langle x_0, x' \rangle - \sqrt{d}\delta \|x'\|}{\|x\|^U \|x'\|} \quad \text{if } \langle x_0, x' \rangle - \sqrt{d}\delta \|x'\| \geq 0 \quad \text{else} \quad \frac{\langle x_0, x' \rangle - \sqrt{d}\delta \|x'\|}{\|x\|^L \|x'\|},$$

$$u^U = \frac{\langle x_0, x' \rangle + \sqrt{d}\delta \|x'\|}{\|x\|^L \|x'\|} \quad \text{if } \langle x_0, x' \rangle + \sqrt{d}\delta \|x'\| \geq 0 \quad \text{else} \quad \frac{\langle x_0, x' \rangle + \sqrt{d}\delta \|x'\|}{\|x\|^U \|x'\|},$$

$$u^U = \min(u^U, 1).$$

where $\Delta \in B_\infty(0, \delta)$. $h(u)$ is a bow shaped function so it is easy to get its interval $[h^L(u), h^U(u)]$. Then we can get the interval of $\Theta(x, x')$, denote as $[\Theta^L(x, x'), \Theta^U(x, x')]$.

$$\Theta^L(x, x') = \frac{\|x\|^L \|x'\|}{2\pi d} h^L(u) \quad \text{if } h^L(u) \geq 0 \quad \text{else} \quad \frac{\|x\|^U \|x'\|}{2\pi d} h^L(u),$$

$$\Theta^U(x, x') = \frac{\|x\|^U \|x'\|}{2\pi d} h^U(u) \quad \text{if } h^U(u) \geq 0 \quad \text{else} \quad \frac{\|x\|^L \|x'\|}{2\pi d} h^U(u).$$

Suppose the $g(x) = \sum_{i=1}^{n} \alpha_i \Theta(x, x_i)$, $\alpha_i$ are known after solving the kernel machine problem. Then we can lower bound and upper bound $g(x)$ as follows.

$$g(x) \geq \sum_{i=1, \alpha_i > 0}^{n} \alpha_i \Theta^L(x, x_i) + \sum_{i=1, \alpha_i < 0}^{n} \alpha_i \Theta^U(x, x_i), \tag{65}$$

$$g(x) \leq \sum_{i=1, \alpha_i < 0}^{n} \alpha_i \Theta^L(x, x_i) + \sum_{i=1, \alpha_i > 0}^{n} \alpha_i \Theta^U(x, x_i). \tag{66}$$

## H.2 IBP for Two-layer Neural Network

See the computation of IBP in [22]. For affine layers of NTK parameterization, the IBP bounds are computed as follows.

$$\mu_{k-1} = \frac{\overline{z}_{k-1} + \underline{z}_{k-1}}{2}$$
$$r_{k-1} = \frac{\overline{z}_{k-1} - \underline{z}_{k-1}}{2}$$
$$\mu_k = \frac{1}{\sqrt{m}} W \mu_{k-1} + b \tag{67}$$
$$r_k = \frac{1}{\sqrt{m}} |W| r_{k-1}$$
$$\underline{z}_k = \mu_k - r_k$$
$$\overline{z}_k = \mu_k + r_k$$

where $m$ is the input dimension of that layer. At initialization, $W$, $\mu_{k-1}$ and $b$ are independent. Since $\mathbb{E}[W] = 0$ and $\mathbb{E}[b] = 0$,

$$\mathbb{E}[\mu_k] = \frac{1}{\sqrt{m}} \mathbb{E}[W] \mathbb{E}[\mu_{k-1}] + \mathbb{E}[b] = 0 \tag{68}$$

Since $|W|$ follows a folded normal distribution (absolute value of normal distribution) and $r_{k-1} \geq 0$, $|W| \geq 0$, $\mathbb{E}[|W|]\mathbb{E}[r_{k-1}] = O(m)$,

$$\mathbb{E}[r_k] = \frac{1}{\sqrt{m}} \mathbb{E}[|W|] \mathbb{E}[r_{k-1}] = O(\sqrt{m}) \tag{69}$$

Thus

$$-\mathbb{E}[\underline{z}_k] = -\mathbb{E}[\mu_k] + \mathbb{E}[r_k] = O(\sqrt{m}) \tag{70}$$
$$\mathbb{E}[\overline{z}_k] = \mathbb{E}[\mu_k] + \mathbb{E}[r_k] = O(\sqrt{m}) \tag{71}$$

And this will cause the robustness lower bound to decrease at a rate of $O(1/\sqrt{m})$. The same results hold for LeCun initialization, which is used in PyTorch for fully connected layers by default.