# OpenReview forum: "On the Equivalence between Neural Network and Support Vector Machine"
_NeurIPS.cc/2021/Conference — NeurIPS 2021 Poster_

### Official Review · Reviewer_vdk1 · 2021-07-15

**Rating:** 6
**Confidence:** 4

**Summary:**

This paper studies ANNs for classification from an NTK perspective. The authors rely on earlier results about the NTK to derive convergence results about ANNs trained with hinge loss (and other loss functions) with an l^2 regularization, showing that these essentially fall in the NTK regime and thus connecting them with SVM. Most of the results are theoretical, but some numerics are also given.

**Ethical Concerns:**

No ethical concerns.

**Limitations And Societal Impact:**

No societal impact.

**Main Review:**

This is an interesting paper, which shows (if the results are correct, of which I am reasonably confident, though not absolutely certain) that neural network classification with hinge loss and with l^2 loss functions fall within the NTK regime, thus allowing us to connect them with SVM. I think the authors should do a better job at emphasizing that it is not trivial that ANNs for classification fall in the NTK regime and connecting with empirical results which seem to show a different behavior (although without an l^2 regularization). The techniques are not particularly new or exciting, but the appendix looks clean.

**Time Spent Reviewing:**

1

---

> ### Author Response · Authors · 2021-08-10
> **Reply to Reviewer vdk1**
>
> Thank you for the positive feedback on our work! We will follow your suggestion to highlight our theoretical contributions.
>
> ### 1. Our contributions
> Although without very hard techniques, we give very meaningful and new theoretical results. Previous theory about NTK cannot handle the case that the loss function has an additional regularization term. They usually assume the loss is a function of the model output without additional regularization terms. Besides, existing theories about NTK usually consider the squared loss which corresponds to a kernel regression (without ridge). The equivalence is only known for ridge regression. As discussed in the footprint and conclusion of [1],  “NTKs can also be used in kernel SVMs, which are not known to be equivalent to training infinitely wide networks. Currently, equivalence is only known for ridge regression”. Thus, the analysis for L2 regularized loss function and the equivalence between NTK SVM and NN are still open and remain to be explored!
>
> In our paper, we give thorough study of regularized loss functions and equivalence with other kernel machines. We not only prove the equivalence between NTK SVM with infinite-width NN for the first time but also generalize to other L2 regularized kernel machines. This is a breakthrough since equivalence is only known for kernel regression by now. Our results can shed light on the understanding of NN from these new equivalent kernel methods (e.g. SVM, kernel ridge regression, SVR, and logistic loss (cross entropy loss)), not just kernel regression. Existing theories about kernel regression and squared loss have limited insights to understand classification problems, since they are usually used for regression problems. Besides, regularization plays an important role in machine learning to restrict the complexity of models. This equivalence between NN and L2 regularized kernel machines may shed light on the understanding of the regularization for NN.
>
>
> Moreover, we proceed to show any finite-width neural network trained by GD is approximately a kernel machine with a data dependent kernel, which may provide lots of insights and applications for NN used in practice. We also reveal the weakness of widely-used existing robustness certification methods based on interval bounds (IBP, Fast-Lin, and convex relaxation based methods). They will give trivial bounds for over-parameterized NN, while we give non-trivial bounds from the perspective of NTK.
>
>
>
> ### 2. Additional application and experiment
> We also conducted additional experiments to show an application of our Theorem 4.1, i.e. computing non-vacuous generalization bound of NN via the corresponding kernel machines. Using theorem 4.1, we can numerically compute the kernel machine that the NN is equivalent to, i.e. we can compute the kernel matrix and the weights at any time during the training. Then one can apply a generalization bound of kernel machines (e.g. Lemma 22 in [2] plus a standard generalization bound using Rademacher complexity, Theorem 3.3 in [3]) to give a generalization bound for this kernel machine (equivalently for this NN). Moreover, we can even apply this generalization bound to optimize NN directly like PAC-Bayes bound [4], which gives NN with guaranteed generalization ability.
>
> Most of the existing generalization bounds of NN [5, 6] are vacacous since they have a dependence on the number of parameters.  Compared to those, the bound for kernel machines does not have a dependence on the number of NN’s parameters. Thus the bound is non-vacuous and this application is promising.
>
> Below is a 2 layer NN with 100 hidden nodes trained on MNIST 2048 data and tested on 2115 data (full test set of 0 and 1). The kernel machine approximates NN very well. And we can get a tight bound of the test loss by computing its Rademacher complexity. We set the confidence parameter $1-\delta=0.99$ in the experiment.
>
> [Computing non-vacuous generalization bound of NN via the corresponding kernel machine.](https://anonymous.4open.science/r/rebuttal-F9F0/output.png) (anonymous link)
>
> In a word, Section 4.1 provides a way to analyze finite-width NN from the perspective of kernel machines. And it may have more insights and applications for finite-width NN used in practice compared to the analysis for infinite-width NN.
>
>
> &nbsp;
> We thank the reviewer for constructive comments. We hope that our answers address all your concerns. We will revise our manuscript based on these comments.
>
>
> ### References
> [1] S. Arora, S. S. Du, Z. Li, R. Salakhutdinov, R. Wang, and D. Yu. Harnessing the power of infinitely wide deep nets on small-data tasks. ICLR 2020.
> [2] Peter L Bartlett and Shahar Mendelson. Rademacher and gaussian complexities: Risk bounds andstructural results.Journal of Machine Learning Research, 3(Nov):463–482, 2002.
> [3] Mohri, Mehryar, Afshin Rostamizadeh, and Ameet Talwalkar. Foundations of machine learning. MIT press, 2018.
> [4] Gintare Karolina Dziugaite and Daniel M Roy. Computing nonvacuous generalization bounds for deep (stochastic) neural networks with many more parameters than training data. UAI, 2017.
> [5] Bartlett P L, Harvey N, Liaw C, et al. Nearly-tight VC-dimension and pseudodimension bounds for piecewise linear neural networks. The Journal of Machine Learning Research, 2019, 20(1): 2285-2301.
> [6] Long P M, Sedghi H. Generalization bounds for deep convolutional neural networks. arXiv preprint arXiv:1905.12600, 2019.

---

### Official Review · Reviewer_fPqE · 2021-07-16

**Rating:** 6
**Confidence:** 3

**Summary:**

This paper theoretically analyzes the connection between Neural Network (NN) and Support Vector Machine (SVM). It shows the equivalence between SVM with Neural tangent Kernel (NTK) and infinite-width NN trained by soft margin loss with subgradient descent.  Besides, they extend their theory to general $\mathit{l}_2$ regularized loss functions and show finite-width NN trained by a $\mathit{l}_2$ regularized loss function is approximately a kernel machine.  It provides robustness certificates for infinite-width NN.

**Limitations And Societal Impact:**

Yes

**Main Review:**

Strength:
1. This paper provides a connection between the SVM and NN  with solid theoretical derivations. It shows that the SVM with NTK and infinite-width NN trained by soft margin loss with subgradient will have the same dynamics and converge to the same solution.
2. The claims of this paper are clear and easy to read.

Weakness:
1. This paper theoretically analyzes the equivalence between SVM with NTK and infinite-width NN trained by soft margin loss with subgradient descent. However, 1-norm soft margin loss is not widely used in NN.

2. The robust analysis is new but not surprising as regularization is widely used to improve the robustness in machine learning. For example, "Regularization Matters: A Nonparametric Perspective on Overparametrized Neural Network" has discussed the ReLU network with $l_2$ regularizer and its connection to the kernel ridge regression.

Questions:
1. In the experiment part, the training dynamics of the network and SVM are similar while the step < $10^2$, but when the number of steps is larger than $10^3$, the dynamic of SVM seems not to agree with NN.

2. I think this paper: "An analytic theory of shallow networks dynamics for hinge loss classification, NeurIPS 2020" might be relevant here. It would be worth discussing the connections.


**Time Spent Reviewing:**

6

---

> ### Author Response · Authors · 2021-08-10
> **Reply to Reviewer fPqE**
>
> Thank you for constructive comments! We hope our answers below address all your concerns.
>
> ### 1. 1-norm soft margin loss is not widely used in NN
> “However, 1-norm soft margin loss is not widely used in NN.”
> - We agree with this comment. However, we have also considered more loss functions, which are more popular and often used in NN, as shown in Section 3.4 and Table 1 at page 6.
> - Our results of SVM provide a theoretical justification for the SVM using NTK as the kernel. More importantly, SVM and 1-norm soft margin loss are just a starting point. We generalize our analysis to general loss functions as shown in Section 3.4 and Table 1. These results show our analysis is also applicable to other frequently used loss functions such as L2 regularized squared loss (which corresponds to kernel ridge regression) and L2 regularized cross entropy loss. And in the experiments of Section 5 and Tabel 3, we have also used L2 regularized squared loss except for the hinge loss.
>
>
> ### 2. Our robustness analysis and (Wang et al., 2020)
> - (Wang et al., 2020) study the **label noise** (the label is generated by a groudth function plus a Gaussian noise) while we consider the robustness of **input perturbation**. The robustness setting is different, i.e. the threat model is different. They study the convergence rate of NN trained by L2 regularized squared loss to an underlying true function, while we give explicit robustness certificates for NNs. Our robustness certificate enables us to compare different models.
> - They consider the setting of two-layer ReLU FC NN with a fixed second layer (very limited setting) trained by L2 regularized squared loss (regression problem) and show it is equivalent to kernel ridge regression (KRR). We consider more general NN architectures (Fully connected NN, CNN, Resnet) and more general loss functions that correspond to more kernel machines (SVM, KRR, etc.). Besides, the robustness of SVM outperforms the kernel ridge regression with the same regularization magnitude a lot, as shown in Table 3 in our paper.
> - It is non-trivial to give robustness certificates for over-parameterized NN. For an over-parameterized neural network, the robustness certificate would be very small (i.e. close to 0, and hence not useful) for widely-used existing robustness certification methods based on interval bounds (IBP, Fast-Lin, and convex relaxation based methods). Our theorem 4.2 can provide non-trivial robustness certificates for such over-parameterized neural networks. It can also be generalized to deeper NNs. This robustness certificate is more useful in practice than the analysis in (Wang et al., 2020).
> - We will cite this paper and add discussion in the revised manuscript.
>
>
> ### Q1. when the number of steps is larger than $10^3$, the dynamic of SVM seems not to agree with NN.
> - This discrepancy is reasonable since the neural network we used in the experiment is not an infinite-width NN. There exists a discrepancy even if the loss function does not have a regularization term, as shown in Figure 3 in [1], which starts to have a discrepancy even before $t=10^1$ (here $t=$step $\times$ learning rate).
> - In Figure 1 (c) of our paper, we also show that this discrepancy will decrease as the width of NN increases.
>
> ### Q2. the connections with "An analytic theory of shallow networks dynamics for hinge loss classification, NeurIPS 2020"
> - Thanks for the related work. We will cite this paper and add discussion in the revised manuscript.
> - They consider one hidden layer NN trained by hinge loss without regularization, while we consider very general NNs trained by regularized hinge loss (and other loss functions). Although we both use hinge loss, note for SVM, it must have a regularization term such that it can achieve max-margin solution.
> - They consider the mean field setting while we focus on the setting of NTK parameterization.
> - They focus on getting insights of the different dynamical regimes of NN and get analytical solutions with a simple linearly separable dataset while we focus on proving the equivalence between infinite-width NN and NTK SVM.
>
>
> &nbsp;
> We thank the reviewer for constructive comments. We hope that our answers address all your concerns. We will revise our manuscript based on these comments.
>
> ### References
> [1] J. Lee, L. Xiao, S. Schoenholz, Y. Bahri, R. Novak, J. Sohl-Dickstein, and J. Pennington. Wide neural networks of any depth evolve as linear models under gradient descent. In Advances in neural information processing systems, pages 8572–8583, 2019.
> [2] Hu T, Wang W, Lin C, et al. Regularization Matters: A Nonparametric Perspective on Overparametrized Neural Network. International Conference on Artificial Intelligence and Statistics. PMLR, 2021: 829-837.

---

> > ### Author Response · Authors · 2021-09-01
> > **Request feedback for our rebuttal**
> >
> > Dear reviewer fPqE,
> >
> > As the discussion period is close to the end and we have not heard back from you, we are reaching out for your feedback. We would love to hear from you regarding our rebuttal response: We have clarified that
> >
> > - our analysis works beyond 1-norm soft margin loss, as shown in Section 3.4 and Table 1 in our paper.
> > - we consider the robustness of input perturbation while (Wang et al., 2020) considers the label noise.
> > - the discrepancy between SVM and NN is reasonable, as also shown in other papers.
> > - the difference between our paper and "An analytic theory of shallow networks dynamics for hinge loss classification, NeurIPS 2020"
> >
> >
> > Your feedback is very important for us to improve the manuscript and we really appreciate your time. Please let us know if you have additional concerns and we are more than happy to discuss and address your concerns.
> >
> > Sincerely,
> > Authors of Paper 1601

---

> > > ### Comment · Reviewer_fPqE · 2021-09-03
> > > **Reply to authors' reponses**
> > >
> > > Thank the authors for the detailed answers to my answers. They have clarified most of my concerns. I would like to increase the score to 6.

---

> > > > ### Author Response · Authors · 2021-09-03
> > > > **Thank you for the updated feedback**
> > > >
> > > > Dear Reviewer fPqE,
> > > >
> > > > Thank you for the new feedback and increasing the score. We are glad that our clarification and the additional details are helpful, and we will include above discussion in the revised manuscript accordingly. Thank you very much for your time and helping us to improve the manuscript!

---

> ### Author Response · Authors · 2021-09-02
> **We would love to hear your feedback on our rebuttal**
>
> Dear Reviewer fPqE,
>
> As the discussion period is close to the end and we have not yet heard back from you, we wanted to reach out to see if our rebuttal response has addressed your concerns.
>
> We are more than happy to discuss further if you have any further concerns and issues, please kindly let us know your feedback. Thank you for your time and help!

---

### Official Review · Reviewer_AUmJ · 2021-07-17

**Rating:** 4
**Confidence:** 2

**Summary:**

The paper shows the equivalence between the infinitely wide NN trained by soft margin loss and the standard 1-norm soft margin SVM with NTK. The authors further consider general loss functions with l2 penalization and show the equivalence between NN and regularized kernel machines. They also provide two practical applications of the developed theory.

**Limitations And Societal Impact:**

The paper lacks theoretical novelty and the application part is not sufficient.

**Main Review:**

The paper establishes the equivalence between neural network and support vector machine and provides several applications according to the developed theory.

My main concerns are as follows:
Some related literature are missing. To name a few, Andras(2002), Ren and Yang (2010), Nong (2012). Without careful comparison and discussion, I cannot see solid contribution based on the existing results.

The paper lacks theoretical novelty. The main results follow from some existing literature and simple analysis.

The application part from the theory lacks further detailed discussion and explanation. The authors need to explain more on Theorem 4.1 and 4.2.

**Time Spent Reviewing:**

6

---

> ### Author Response · Authors · 2021-08-10
> **Reply to Reviewer AUmJ**
>
> Thank you for constructive comments! We hope our answers below address all your concerns.
>
> ### 1. Some related literature are missing
> Thanks for the related literature. The main difference between our work and Andras (2002), Ren and Yang (2010) is that they consider specific kinds of neural networks (regularization neural network and RBF neural network), which **have very similar structures with that of kernel machines**, making it relatively easy to establish a connection with kernel machines. And these kinds of NNs are not widely used currently. We consider NNs widely used in practice (e.g. multi-layer fully connected NN, CNN, Resnet), which do not have such structures that are similar to kernel machines. Thus it is harder to establish a connection with kernel machines. Please see the detailed comparison below. We will cite these papers and add the discussion in the revised manuscript.
>
> Andras P. The equivalence of support vector machine and regularization neural networks. Neural Processing Letters, 2002, 15(2): 97-104.
> - They consider one kind of specific NN, regularization neural network [1, 2]. It is a one-hidden layer NN using Green’s function $G(x, x^i)$ as the activation function of the hidden layer, whose solution has a form of $g(x) = \sum_{i=1}^n w_i G(x, x^i)$ (see e.g. Eq. (7) and Fig. 1 in [1]). This form of solution is very similar to that of kernel machines, so it is easy to connect them together. For example, given a regularization neural network of form $g(x) = \sum_{i=1}^n w_i G(x, x^i)$, there exists an equivalent problem of SVM by using $G(x, x^i)$ as the kernel function (Andras(2002) and [2]). This is trivial equivalence. Besides, this kind of NN is different from what we use in practice, which does not have such a convenient form of solution. Thus it is harder to connect them with kernel methods. Our results hold for general NNs (multilayer fully connected NN, CNN, Resnet) used today.
>
> - While they do not consider the optimization, we consider the optimization by (sub)gradient descent. We show even the dynamics of (sub)gradient descent of the infinite-width NN and NTK SVM are the same and they converge to the same solution.
>
>
> Jinxia R, Sai Y. RBF Neural Networks Optimization Algorithm Based on Support Vector Machine and Its Application. 2010 2nd International Conference on Information Engineering and Computer Science. 2010.
> - They consider the RBF neural network. Radial basis function network is a one hidden layer neural network that uses radial basis functions as activation functions (see e.g. wiki of Radial basis function network), which is different from the NN used today. Since radial basis function has almost the same form as Gaussian kernel function, it is quite easy to connect RBF neural network with SVM using Gaussian kernel.
> - They propose to use a genetic algorithm to optimize the parameters in SVM then use the solution of SVM to construct a RBF neural network. We do not see any difference between this method and directly using the SVM.
>
>
> Nong J. Radial Basis Function Neural Networks Optimization Algorithm Based on SVM. Communications and Information Processing. Springer, Berlin, Heidelberg, 2012: 291-298.
> - This paper has exactly the same content with the second paper (even the figures).
>
> ### 2. Theoretical novelty
> Although without very hard techniques, we give very meaningful and new theoretical results.
>
> Previous theory about NTK cannot handle the case that the loss function has an additional regularization term. They usually assume the loss is a function of the model output without additional regularization terms. Please also see a detailed comparison in the reply to the Reviewer w1TB. Besides, existing theories about NTK usually consider the squared loss which corresponds to a kernel regression (without ridge). The equivalence is only known for ridge regression. As discussed  in the footprint and conclusion of [3],  “NTKs can also be used in kernel SVMs, which are not known to be equivalent to training infinitely wide networks. Currently, equivalence is only known for ridge regression”. Thus, the analysis for L2 regularized loss function and the equivalence between NTK SVM and NN are still open and remain to be explored!
>
> In our paper, we give thorough study of regularized loss functions and equivalence with other kernel machines. We not only prove the equivalence between NTK SVM with infinite-width NN for the first time but also generalize to other L2 regularized kernel machines. This is a breakthrough since equivalence is only known for kernel regression by now. Our results can shed light on the understanding of NN from these new equivalent kernel methods (e.g. SVM, kernel ridge regression, SVR, and logistic loss (cross entropy loss)), not just kernel regression. Existing theories about kernel regression and squared loss have limited insights to understand classification problems, since they are usually used for regression problems. Besides, regularization plays an important role in machine learning to restrict the complexity of models. This equivalence between NN and L2 regularized kernel machines may shed light on the understanding of the regularization for NN.
>
>
> Moreover, we proceed to show any finite-width neural network trained by GD is approximately a kernel machine with a data dependent kernel, which may provide lots of insights and applications for NN used in practice (see our reply for the third concern). We also reveal the weakness of widely-used existing robustness certification methods based on interval bounds (IBP, Fast-Lin, and convex relaxation based method). They will give trivial bounds for over-parameterized NN (see our explanation for the third concern), while we give non-trivial bounds from the perspective of NTK.
>
> ### 3. Applications of our theoretical results
>
> Theorem 4.1 shows that any finite-width neural network trained by GD is approximately a kernel machine with a data dependent kernel. Theorem 4.1 provides lots of potential insights and applications. Here we give an application that computes non-vacuous generalization bound of NN via the corresponding kernel machines: Using theorem 4.1, we can numerically compute the kernel machine that the NN is equivalent to, i.e. we can compute the kernel matrix and the weights at any time during the training. Then one can apply a generalization bound of kernel machines (e.g. Lemma 22 in [4] plus a standard generalization bound using Rademacher complexity, Theorem 3.3 in [5]) to give an generalization bound for this kernel machine (equivalently for this NN). Moreover, we can even apply this generalization bound to optimize NN directly like PAC-Bayes bound [6], which gives NN with guaranteed generalization ability.
>
> Most of the existing generalization bounds of NN [7, 8] are vacacous since they have a dependence on the number of parameters. Compared to those, the bound for kernel machines does not have a dependence on the number of NN’s parameters. Thus the bound is non-vacuous and this application is promising.
>
> Below is a 2 layer NN with 100 hidden nodes trained on MNIST 2048 data and tested on 2115 data (full test set of 0 and 1). The kernel machine approximates NN very well. And we can get a tight bound of the test loss by computing its Rademacher complexity. We set the confidence parameter $1-\delta=0.99$ in the experiment.
>
> [Computing non-vacuous generalization bound of NN via the corresponding kernel machine.](https://anonymous.4open.science/r/rebuttal-F9F0/output.png) (anonymous link)
>
> In a word, Section 4.1 provides a way to analyze finite-width NN from the perspective of kernel machines. And it may have more insights and applications for finite-width NN used in practice compared to the analysis for infinite-width NN.
>
> Theorem 4.2 gives a *non-trivial* robustness certificate for a two-layer ReLU NN. For an over-parameterized neural network, the robustness certificate would be very small (i.e. close to 0, and hence not useful) for widely-used existing robustness certification methods based on interval bounds (IBP, Fast-Lin, and convex relaxation based method). The reason is because when we increase the width $m$ of NN, the interval of the output would increase at a rate of $O(\sqrt{m})$, which will cause the robustness bound decrease at a rate of $O(1 / \sqrt{m})$ (see our analysis in Appendix F.2). Fortunately, with the results in our paper, we could obtain *non-trivial* robustness certificate for overparameterized NN, as we demonstrated in Section 4.2 and the experiments in Section 5.
>
>
> &nbsp;
> We thank the reviewer for constructive comments. We hope that our answers address all your concerns. We will revise our manuscript based on these comments.
>
>
> ### References
> [1] Poggio T, Girosi F. Networks for approximation and learning. Proceedings of the IEEE, 1990, 78(9): 1481-1497.
> [2] Smola A J, Schölkopf B, Müller K R. The connection between regularization operators and support vector kernels. Neural networks, 1998, 11(4): 637-649.
> [3] S. Arora, S. S. Du, Z. Li, R. Salakhutdinov, R. Wang, and D. Yu. Harnessing the power of infinitely wide deep nets on small-data tasks. ICLR 2020.
> [4] Peter L Bartlett and Shahar Mendelson. Rademacher and gaussian complexities: Risk bounds andstructural results.Journal of Machine Learning Research, 3(Nov):463–482, 2002.
> [5] Mohri, Mehryar, Afshin Rostamizadeh, and Ameet Talwalkar. Foundations of machine learning. MIT press, 2018.
> [6] Gintare Karolina Dziugaite and Daniel M Roy. Computing nonvacuous generalization bounds for deep (stochastic) neural networks with many more parameters than training data. UAI, 2017.
> [7] Bartlett P L, Harvey N, Liaw C, et al. Nearly-tight VC-dimension and pseudodimension bounds for piecewise linear neural networks. The Journal of Machine Learning Research, 2019, 20(1): 2285-2301.
> [8] Long P M, Sedghi H. Generalization bounds for deep convolutional neural networks. arXiv preprint arXiv:1905.12600, 2019.

---

> > ### Author Response · Authors · 2021-09-01
> > **Request feedback for our rebuttal**
> >
> > Dear reviewer AUmJ,
> >
> > As the discussion period is close to the end and we have not heard back from you, we are reaching out for your feedback. We would love to hear from you regarding our rebuttal response:
> >
> > - We give a detailed comparison between our work and the related papers you provide. These papers are not quite relevant to our paper since they consider specific neural networks that have very similar structures with kernel machines and are not widely used in practice.
> > - We explain more details about our contributions compared with previous results. Our paper studies the regularized loss functions (which previous theory cannot handle) and establishes the equivalence between infinite-width NN with kernel machines beyond kernel regression for the first time.
> > - We give a more detailed discussion and explanation for Theorem 4.1 and 4.2. We show novel applications of these theories to give non-trivial robustness certificates for over-parameterized NNs and compute non-vacuous generalization bound of NN via the corresponding kernel machines.
> >
> >
> > Your feedback is very important for us to improve the manuscript and we really appreciate your time. Please let us know if you have additional concerns and we are more than happy to discuss and address your concerns.
> >
> > Sincerely,
> > Authors of Paper 1601

---

> > ### Author Response · Authors · 2021-09-02
> > **We would love to have your feedback on our rebuttal response**
> >
> > Dear Reviewer AUmJ,
> >
> > As the discussion period is close to the end and we have not yet heard back from you, we wanted to reach out to see if our rebuttal response has addressed your concerns.
> >
> > We are more than happy to discuss if you have any further concerns and issues, please kindly let us know your feedback. Thank you for your time and help!

---

### Official Review · Reviewer_w1TB · 2021-07-17

**Rating:** 5
**Confidence:** 4

**Summary:**

In this paper, the authors consider the infinite-width limit of neural networks and show the equivalence of NN trained by soft margin loss and SVM with neural tangent kernel. They proceed to show that any finite-width neural networks trained by GD is approximately a kernel machine with a certain data dependent kernel. They further show that one can get a robustness certificate from NTK.

**Limitations And Societal Impact:**

Yes

**Main Review:**

This work belongs to the recent push in the deep learning theory community that showed that in some large-width limit, NNs effectively behave as kernel methods, with the so called neural neural tangent kernel. However I have several concerns:

- While it is true that a lot of attention has been given to the linear regime in the past three years, the limitations of this approach have been studied extensively since then. Even though there was a surge of interest for kernel methods, which were proven more efficient than previously thought, it is now clear that the NTK regime provides only limited insights into the behavior of neural networks.

- While I am not sure whether the hinge-loss, and non-smooth loss have been explicitly studied before, plenty of work have already shown quite generally that large width network behave as kernel methods, for example Chizat, Oyallon and Bach 2018 (especially, here no rates of convergence are provided so one can take a sequence of smooth approximation).

- I am not sure I understand section 4.1: while it is true that you can write it as a solution of a kernel method, the kernel is now data dependent. It seems to be a trivial statement. For example, it is always true that one can train a neural network and then say that this is the solution of a kernel methods by retraining the last layer while keeping the other layers fixed with the learned weights. This does not give any particular insight, except that gradient descent is a way to learn a good kernel.

- I understand the big improvement in robustness compared to IBP, but is it not a weakness of IBP? Will the bound of Theorem 4.2 work well when using the kernel obtained at the end of the learning trajectory?

For all the reasons presented above, I don’t think this paper as currently written is suitable for NeurIPS.


--- after rebuttal ---
Changed rating to 5.


**Time Spent Reviewing:**

2

---

> ### Author Response · Authors · 2021-08-10
> **Reply to Reviewer w1TB**
>
> Thank you for constructive comments! We hope our answers below address all your concerns.
>
> ### 1. NTK regime provides only limited insights
> We acknowledge the limitations of linear regime. But it still gives good approximations for over-parameterized NNs [1]. Besides, it can inspire powerful applications in deep learning, such as pruning [2], neural architecture search [3], graph neural networks [4] and matrix completion [5]. Our new theoretical results would give more insights for NN and inspire more applications.
>
> Moreover, our Section 4.1 is for finite-width NN, which gives insights and applications for NN in nonlinear regime (see our reply for the third concern).
>
>
> ### 2. Plenty of work already shown quite generally that large width network behave as kernel methods, e.g. Chizat, Oyallon and Bach 2018
>
> Previous results cannot handle the case that the loss function has an additional regularization term. For example,
> - (Chizat, Oyallon and Bach 2018) mainly talk about lazy training and linearization of neural networks but **do not make direct connection with any kernel method**. Besides, they assume the loss is a function of model output, which **does not include the case of regularization**. This can also be seen from their derivation, in which they only consider the derivative of the loss function w.r.t the model output but not include other terms (e.g. regularization). They consider squared loss as a special case.
> - [7] also assume the loss is a function of model output, which **does not include the case of regularization**. They also consider squared loss as a special case, which they show is equivalent to kernel ridge regression (with zero ridge).
> - [1, 8] consider the squared loss, which **does not include the case of regularization**.
> - [9] study the generalization of two-layer NN with squared loss, which **does not include the case of regularization**. The results provide limited insights for classification problems since squared loss is usually used for regression problems.
>
> Above discussion of literature shows that the regularized loss functions have not been studied well yet. Besides, existing theories about NTK usually consider the squared loss which corresponds to a kernel regression (without ridge). The equivalence is only known for ridge regression. As discussed in [10] in their footprint and conclusion,  “NTKs can also be used in kernel SVMs, which are not known to be equivalent to training infinitely wide networks. Currently, equivalence is only known for ridge regression”. Thus, the analysis for L2 regularized loss function and the equivalence between NTK SVM and NN are non-trivial and remain to be explored!
>
> In our paper, we give thorough study of regularized loss functions and equivalence with other kernel machines. We not only prove the equivalence between NTK SVM with infinite-width NN for the first time but also generalize to other L2 regularized kernel machines. This is a breakthrough since equivalence is only known for kernel regression by now. Our results can shed light on the understanding of NN from these new equivalent kernel methods (e.g. SVM, kernel ridge regression, SVR, and logistic loss (cross entropy loss)), not just kernel regression. Existing theories about kernel regression and squared loss have limited insights to understand classification problems, since they are usually used for regression problems. Besides, regularization plays an important role in machine learning to restrict the complexity of models. This equivalence between NN and L2 regularized kernel machines may shed light on the understanding of the regularization for NN.
>
>
>
> ### 3. Sec 4.1 seems trivial and does not give any particular insight
>
> “For example, it is always true that one can train a neural network and then say that this is the solution of a kernel method by retraining the last layer while keeping the other layers fixed with the learned weights”.
>
> This is not always true. While NN is linear for the last layer, if we fix the other layers (the layers before the last layer) and see the output before the last layer as a feature mapping $\phi(x)$, then the NN is learning a linear function in a feature space. But a linear function in some high dimensional space does not make it a kernel method. Kernel method requires that the solution has a form of $g(x) = \sum_i^n \alpha_i k(x, x_i)$. This usually has a requirement for the loss function. See e.g. Representer theorem.
>
> Besides, this case of fixing the layers is a quite limited special case and is different from a fully-trained NN (the usual practice). And this case can also be seen as a special case of our fully trained neural network, since we can always set the gradients of the fixed parameters as 0 in the tangent kernel as they did in [10]. As we also discussed at line 191 in the supplementary material, if we only train the last layer of the infinite-width NN,  it corresponds to an SVM with a Neural Network Gaussian Process (NNGP) kernel [11]. And the NNGP kernel can also be seen as a special case of NTK when only the last layer of NN is trained. But these are really just special cases of our analysis.
>
> &nbsp;
> “This does not give any particular insight, except that gradient descent is a way to learn a good kernel.”
>
> Section 4.1 is nontrivial and provides lots of potential insights and applications. Here we give an application that computes the non-vacuous generalization bound of NN via the corresponding kernel machines. Using theorem 4.1, we can numerically compute the kernel machine that the NN is equivalent to, i.e. we can compute the kernel matrix and the weights at any time during the training. Then one can apply a generalization bound of kernel machines (e.g. Lemma 22 in [12] plus a standard generalization bound using Rademacher complexity, Theorem 3.3 in [13]) to give an generalization bound for this kernel machine (equivalently for this NN). Moreover, we can even apply this generalization bound to optimize NN directly like PAC-Bayes bound [14], which gives NN with guaranteed generalization ability.
>
> Most of the existing generalization bounds of NN [15, 16] are vacacous since they have a dependence on the number of parameters.  Compared to those, the bound for kernel machines does not have a dependence on the number of NN’s parameters. Thus the bound is non-vacuous and this application is promising.
>
> Below is a 2 layer NN with 100 hidden nodes trained on MNIST 2048 data and tested on 2115 data (full test set of 0 and 1). The kernel machine approximates NN very well. And we can get a tight bound of the test loss by computing its Rademacher complexity. We set the confidence parameter $1-\delta=0.99$ in the experiment.
>
> [Computing non-vacuous generalization bound of NN via the corresponding kernel machine.](https://anonymous.4open.science/r/rebuttal-F9F0/output.png) (anonymous link)
>
> In a word, Section 4.1 provides a way to analyze finite-width NN from the perspective of kernel machines. And it may have more insights and applications for finite-width NN used in practice compared to the analysis for infinite-width NN.

---

> > ### Author Response · Authors · 2021-08-10
> > **Reply to Reviewer w1TB (continued)**
> >
> > ### 4. Theorem 4.2 and IBP bounds
> >
> > “I understand the big improvement in robustness compared to IBP, but is it not a weakness of IBP?”
> >
> > This is the weakness of all interval-based robustness verification methods. For an over-parameterized neural network, the robustness certificate would be very small (i.e. close to 0, and hence not useful) for widely-used existing robustness certification methods based on interval bounds (IBP, Fast-Lin, and convex relaxation based method). The reason is because when we increase the width $m$ of NN, the interval of the output would increase at a rate of $O(\sqrt{m})$, which will cause the robustness bound decrease at a rate of $O(1 / \sqrt{m})$ (see our analysis in Appendix F.2). Fortunately, with our results in the paper, we could obtain *non-trivial* robustness certificate for overparameterized NN, as we demonstrated in Section 4.2 and the experiments in Section 5.
> >
> > “Will the bound of Theorem 4.2 work well when using the kernel obtained at the end of the learning trajectory?”
> >
> > Yes, the bound of Theorem 4.2 would still work with the kernel obtained at the end of learning trajectory because the NTK of infinite-width NN would stay constant during training as shown in [7] and  [17]
> >
> > &nbsp;
> >
> > We thank the reviewer for constructive comments. We hope that our answers address all your concerns. We will revise our manuscript based on these comments.
> >
> >
> > ### References
> > [1] J. Lee, L. Xiao, S. Schoenholz, Y. Bahri, R. Novak, J. Sohl-Dickstein, and J. Pennington. Wide neural networks of any depth evolve as linear models under gradient descent. In Advances in neural information processing systems, pages 8572–8583, 2019.
> > [2] Wang C, Zhang G, Grosse R. Picking Winning Tickets Before Training by Preserving Gradient Flow. ICLR 2019.
> > [3] Chen W, Gong X, Wang Z. Neural Architecture Search on ImageNet in Four GPU Hours: A Theoretically Inspired Perspective. ICLR 2020.
> > [4] Du S S, Hou K, Salakhutdinov R R, et al. Graph neural tangent kernel: Fusing graph neural networks with graph kernels. Advances in Neural Information Processing Systems, 2019, 32: 5723-5733.
> > [5] Radhakrishnan A, Stefanakis G, Belkin M, et al. Simple, Fast, and Flexible Framework for Matrix Completion with Infinite Width Neural Networks. arXiv preprint arXiv:2108.00131, 2021.
> > [6] Chizat, Lenaic and Oyallon, Edouard and Bach, Francis. On Lazy Training in Differentiable Programming. arXiv preprint arXiv:1812.07956.
> > [7] A. Jacot, F. Gabriel, and C. Hongler. Neural tangent kernel: Convergence and generalization in neural networks. In Advances in neural information processing systems, pages 8571–8580, 2018.
> > [8] S. Arora, S. S. Du, W. Hu, Z. Li, R. R. Salakhutdinov, and R. Wang. On exact computation with an infinitely wide neural net. In Advances in Neural Information Processing Systems, pages 8141–8150, 2019.
> > [9] S. Arora, S. S. Du, W. Hu, Z. Li, and R. Wang. Fine-grained analysis of optimization and gener- alization for overparameterized two-layer neural networks. arXiv preprint arXiv:1901.08584, 2019.
> > [10] S. Arora, S. S. Du, Z. Li, R. Salakhutdinov, R. Wang, and D. Yu. Harnessing the power of infinitely wide deep nets on small-data tasks. ICLR 2020.
> > [11] Jaehoon Lee, Jascha Sohl-dickstein, Jeffrey Pennington, Roman Novak, Sam Schoenholz, and Yasaman Bahri. Deep neural networks as gaussian processes. In International Conference on Learning Representations, 2018.
> > [12] Peter L Bartlett and Shahar Mendelson. Rademacher and gaussian complexities: Risk bounds andstructural results.Journal of Machine Learning Research, 3(Nov):463–482, 2002.
> > [13] Mohri, Mehryar, Afshin Rostamizadeh, and Ameet Talwalkar. Foundations of machine learning. MIT press, 2018.
> > [14] Gintare Karolina Dziugaite and Daniel M Roy. Computing nonvacuous generalization bounds for deep (stochastic) neural networks with many more parameters than training data. UAI, 2017.
> > [15] Bartlett P L, Harvey N, Liaw C, et al. Nearly-tight VC-dimension and pseudodimension bounds for piecewise linear neural networks. The Journal of Machine Learning Research, 2019, 20(1): 2285-2301.
> > [16] Long P M, Sedghi H. Generalization bounds for deep convolutional neural networks. arXiv preprint arXiv:1905.12600, 2019.
> > [17] C. Liu, L. Zhu, and M. Belkin. On the linearity of large non-linear models: when and why the tangent kernel is constant. Advances in Neural Information Processing Systems, 33, 2020.

---

> > > ### Author Response · Authors · 2021-09-01
> > > **Request feedback for our rebuttal**
> > >
> > > Dear reviewer w1TB,
> > >
> > > As the discussion period is close to the end and we have not heard back from you, we are reaching out for your feedback. We would love to hear from you regarding our rebuttal response:
> > >
> > > - We have provided a detailed comparison between our work and previous works to show our contributions more clearly. In particular, we **solved an open problem about the equivalence between neural networks and SVM (and other kernel machines for classification and regression), which is the equivalence beyond ridge regression (usually for regression) for the first time**.
> > > - We also gave additional applications for Theorem 4.1 (computing non-vacuous generalization bound of NN via the corresponding kernel machine).
> > > - We clarified the weakness of IBP bounds and the concerns about Theorem 4.2.
> > >
> > > Your feedback is very important for us to improve the manuscript and we really appreciate your time. Please let us know if you have additional concerns and we are more than happy to discuss and address your concerns.
> > >
> > > Sincerely,
> > > Authors of Paper 1601

---

> > > > ### Comment · Reviewer_w1TB · 2021-09-01
> > > > **Feedback**
> > > >
> > > > Dear authors,
> > > >
> > > > Thank you for this very detailed rebuttal, that led me to revise my opinion quite a bit. On the different points:
> > > >
> > > > 1. NTK regime provides only limited insights:
> > > >
> > > > I think NTK does provide a lot of insights and is an important topic of research. However, a lot of work has been published on it and I think that new papers should be justified and provide either important new conceptual insights or be very general, rather than showing NTK regime for a new model.
> > > >
> > > > 2. Plenty of work already shown quite generally that large width network behave as kernel methods, e.g. Chizat, Oyallon and Bach 2018.
> > > >
> > > > Some of the work cited provide sufficient conditions for the prediction model to stay closed to its linearization. Then to show the correspondence with NTK, it suffices to take the number of neurons to infinity... However, I changed my opinion about the contribution of adding an $\ell_2$ regularization. In several of the references, it is suggested that `weight decay' ($\ell_2$ regularization) would lead to a significant change of the weights and will break the linear/kernel regimes, while from Liu et al, 2020, it is clear that what is important is the constancy of the kernel (very small Hessian along the trajectory). This does not seem to be well understood. Furthermore, going from the constancy of the kernel, to the same dynamics between large NNs and SVM with NTK is less obvious than I first thought.
> > > > However, even if it is somewhat conceptually interesting, I think that it is sufficiently simple technically to require a more interesting analysis, for example non-asymptotic similar to Theorem 5.4 in https://arxiv.org/pdf/2103.09177.pdf.
> > > >
> > > >
> > > > 3. Sec 4.1 seems trivial and does not give any particular insight
> > > >
> > > > I want to clarify one technical point. There are two ways of introducing a RKHS: through a kernel $K(\cdot, \cdot)$ or through a feature map $\phi : X \to F$, where $F$ is a general Hilbert space. The two constructions are equivalent with $K(x,y) = \< \phi (x) , \phi (y) \>_F$ and any function in the RKHS can be written as $f (x) = \< \theta , \phi (x) \>_F$ with $\theta \in F$. For any linear model $f(x) = \theta^T \phi(x)$, with map $\phi(x) \in \mathbb{R}^m$, this can be seen as kernel method with finite rank kernel $K(x,y) = \phi(x)^T \phi(y)$. It is an easy exercise to show that when $\hat \theta$ is the solution of a ERM, then one can rewrite $f( x) = \sum_i b_i \phi(x_i)^T \phi(x)$ (by duality, representer theorem etc).
> > > >
> > > > I think there are several ways of writing a function as the solution of a time dependent kernel method... I think this is a non surprising statement to make, but as the authors defended in the rebuttal, this might lead to non-obvious applications like robustness certificates.
> > > >
> > > >
> > > > For the reasons mentioned above, I am raising my rating to 5. I think the paper should be published somewhere, but I find it a bit `light' compared to the other papers I have been reviewing for NeurIPS.

---

> > > > > ### Author Response · Authors · 2021-09-02
> > > > > **Reply to Reviewer w1TB**
> > > > >
> > > > > Dear reviewer w1TB,
> > > > >
> > > > > Thank you for raising the rating and constructive comments! We hope our answers below address your concerns.
> > > > >
> > > > > ### 1. new papers should be justified and provide either important new conceptual insights or be very general, rather than showing NTK regime for a new model.
> > > > >
> > > > > Our results are very general since our analysis holds not just for SVM but for general loss functions, as shown in Section 3.4 and Table 1. When the coefficient of the regularization term is 0, we recover the previous results. SVM is just a starting point and our analysis does not have specific dependence on the loss function except differentiable.
> > > > >
> > > > > ### 2. non-asymptotic analysis
> > > > >
> > > > > Thank you for acknowledging our contributions and the good advice. Yes, with similar assumptions of the singular values and Lipschitz constant, we can get similar bounds as Theorem 5.1 and 5.4 in https://arxiv.org/pdf/2103.09177.pdf. Since we already have similar dynamics of SVM and NN, the only thing left is to bound the difference between them using these assumptions. For the convergence, we already give a linear convergence rate for the gradient flow of SVM in Appendix B.1, although we did not write it in the main paper.
> > > > >
> > > > > ### 3. this might lead to non-obvious applications like robustness certificates
> > > > >
> > > > > Yes, our robustness certificates and computing generalization bound of NN via the corresponding kernel machine are novel and promising. Through these applications, we could obtain *non-trivial* robustness certificates for overparameterized NNs compared with existing interval-based robustness verification methods. We could get *non-vacuous* generalization bounds for finite-width NN compared with existing generalization bounds for NN.
> > > > >
> > > > > Sincerely,
> > > > > Authors of Paper 1601

---

> > > > > > ### Author Response · Authors · 2021-09-02
> > > > > > **We have responded to your new feedback, please let us know if you have further concerns**
> > > > > >
> > > > > > Dear reviewer w1TB,
> > > > > >
> > > > > > Based on the above clarification on #1 generality of our work, #2 non-asymptotic analysis, and #3 novelty of the applications, we hope that we have clarified the contributions of our work are non-trivial and substantial. We will add our above discussions in the revised manuscript, and we hope you will reconsider the rating.  Thank you again for helping us improve our work!
> > > > > >
> > > > > > Sincerely,
> > > > > > Authors of Paper 1601

---

### Decision · Program_Chairs · 2021-09-27

**Decision:**

Accept (Poster)

**Comment:**

This paper establishes the equivalence between neural network and support vector machines, following the recent trend in identifying equivalence between infinite-width neural networks and kernel machines. In particular, neural network classification trained by soft margin loss with subgradient descent and extension to general regularized loss functions (including L2 regularization) and show infinite-width limit falls in the NTK regime. As applications of the developed theory, authors obtain non-vacuous generalization bound of NN via the corresponding kernel machines as well as robustness certificates.

After the discussion, the reviewer’s final recommendations are 2 weak accepts, 1 weak reject  and 1 reject. Given that the reject recommendation comes with low confidence and was disengaged despite the author's significant attempt to clarify, the AC is inclined to significantly downweight the suggestion. Even so the paper remains borderline.

In the end, the majority of reviewers agreed there’s merit in the paper (new equivalence, L2 regularization, application to robustness and generalization bound) and should be published. The question remained whether NeurIPS is the appropriate venue. Given the paper tackles an important question, providing non-trivial insights and equivalence they established are deemed to be correct to the expert reviewers, the AC believes the paper’s claim is significant enough and would benefit NeurIPS participants.

As a side: One of the reviewers in the end had concern/question on the generalization bound plots (provided in anonymous link). It appears that on the right panel the train/test curves follow the same line which is questionable. Also the experiment should be compared to much further in the training time (near convergence).

---

> ### Public Comment · Authors · 2021-11-10
> **Thank you for the final comments, some new results are updated in the manuscript**
>
> Dear area chair and reviewers,
>
> Thank you for positive feedback and accepting our paper. Based on the suggestions from the reviewers, we provide some new results in our revised paper, including the convergence rate of NN under some assumption in Theorem 3.3 and finite-width bounds between NN and corresponding kernel machines in Theorem 3.5. These new theoretical results may be of independent interests.
>
> For the questions pointed out,
> -  The train/test curves follow the same line which is questionable
>
> First, the train and test loss curves do not follow the same line. If taking a careful look, they have some differences (the difference is in the magnitude of $10^{-3}$). The train and test loss curves are very similar because the task is binary classification which is a simple task and thus the averaged training and test loss are very similar. Our codes for the experiments are available at https://github.com/leslie-CH/equiv-nn-svm
>
>
> - The experiment should be compared to much further in the training time (near convergence)
>
> We did not compare them near the convergence because the kernel machine does not approximate NN very well when trained for a long time. This is because Theorem 4.1 is based on the assumption of gradient flow. This approximation error may be improved by using some professional ODE solvers or improving the theorem to gradient descent version.
>
>
>
> Sincerely,
> Authors of Paper 1601

---

> > ### Public Comment · Authors · 2022-07-08
> > **Updated manuscript and remark on Theorem 4.1**
> >
> > Dear readers,
> >
> > Thank you for your interest in our work. We have updated some typos, related literature on **implicit bias of neural network towards the max-margin solution**, and one **remark on Theorem 4.1** in the latest ArXiv version.
> >
> > The remark is to note that the kernel in Theorem 4.1 is valid only when $|l’(f_t(x_i), y_i)|$ is a constant, e.g. $l’(f_t(x_i), y_i) = -y_i$ at the initial training stage of hinge loss with $f_0(x) = 0$, otherwise the kernel is not symmetric.
> > Our updated manuscript is at https://arxiv.org/abs/2111.06063.
> >
> > Sincerely,
> > Authors of Paper 1601